# Deep Unlearn: Benchmarking Machine Unlearning

## Abstract

Machine unlearning (MU) aims to remove the influence of particular data points from the learnable parameters of a trained machine learning model. This is a key capability in light of data privacy requirements, trustworthiness, and safety in deployed models. MU is particularly challenging for deep neural networks (DNNs), such as convolutional nets or vision transformers, as such DNNs tend to memorize a notable portion of their training dataset. Nevertheless, the community lacks a rigorous and multifaceted study that looks into the success of MU methods for DNNs. In this paper, we investigate 18 state-of-the-art MU methods across various benchmark datasets and models, with each evaluation conducted over 10 different initializations, a comprehensive evaluation involving MU over 100K models. We show that, with the proper hyperparameters, Masked Small Gradients (MSG) and Convolution Transpose (CT), consistently perform better in terms of model accuracy and run-time efficiency across different models, datasets, and initializations, assessed by population-based membership inference attacks (MIA) and per-sample unlearning likelihood ratio attacks (U-LiRA). Furthermore, our benchmark highlights the fact that comparing a MU method only with commonly used baselines, such as Gradient Ascent (GA) or Successive Random Relabeling (SRL), is inadequate, and we need better baselines like Negative Gradient Plus (NG+) with proper hyperparameter selection.

## 1 Introduction

Machine unlearning aims to remove the influence of a specified subset of training data points from trained models [10]. This process is crucial for enhancing privacy preservation, model safety, and overall model quality. MU helps ensure compliance with the right to be forgotten [23], removes erroneous data points that negatively impact model performance [43], and eliminates biases introduced by parts of the training data [14]. Deep neural networks present significant challenges for MU due to their computationally intensive training requirements, highly non-convex loss landscapes, and tendency to memorize substantial portions of their training data [22, 12, 21]. The key open challenges in MU include: (1) A degradation in model accuracy often accompanies unlearning; (2) Some MU methods require the model to be trained in specific ways, such as saving checkpoints, tracking accumulated gradients, or training with differential privacy, limiting their applicability to already deployed models; (3) Assurance of information removal is difficult as there are no reliable metrics to measure it accurately; (4) There is no consensus on which methods are the most effective.

A branch of MU known as *exact unlearning* aims to guarantee that the specified *forget* data have been completely removed from the model. The most reliable exact MU method is to *retrain* the model from scratch while excluding the forget data. Another exact MU that offers data removal guarantees is SISA (Sharded, Isolated, Sliced, Aggregated) [9]. However, exact MU is computationally prohibitive, emphasizing the need for more efficient methods. The alternative branch is *approximate unlearning*

Submitted to the 38th Conference on Neural Information Processing Systems (NeurIPS 2024) Track on Datasets and Benchmarks. Do not distribute.

that aims to approximate data deletion and is often less precise, but more computationally efficient. Approximate MU lacks theoretical guarantees, necessitating empirical evaluations to determine their effectiveness, reliability, and computational efficiency across various datasets. Thus, the community needs a proper benchmarking and evaluation of state-of-the-art approximate MU methods.

In this paper, we benchmark 18 state-of-the-art approximate MU methods across 5 datasets and 2 DNN architectures commonly used in computer vision: ResNet18 [32] and TinyViT [48]. The 18 methods consist of 3 classical baseline methods Fine-tuning (FT), Gradient Ascent (GA), Successive Random Labels (SRL); 7 high-ranking methods in the NeurIPS'2023 Machine Unlearning competition [45], we name them Forget-Contrast-Strengthen (FCS), Masked-Small-Gradients (MSG), Confuse-Finetune-Weaken (CFW), Prune-Reinitialize-Match-Quantize (PRMQ), Convolution-Transpose (CT), Knowledge-Distillation-Entropy (KDE) and Repeated-Noise-Injection (RNI); and 8 recently published papers: Saliency Unlearning (SalUN) [20], Catastrophic Forgetting-K (CF-k) [24], Exact Unlearning-K (EU-k) [24], SCalable Remembering and Unlearning unBound (SCRUB) [37], Bad Teacher (BT) [16], Fisher Forgetting (FF) [25], Influence Unlearning (IU) [35, 33] and Negative Gradient Plus (NG+) [37]. We evaluate the different unlearning methods in terms of four major aspects: privacy evaluation, performance retention, computational efficiency, and unlearning reliability.

The **contributions** of our study is to address the following research questions:

*Q1. Are the commonly used MU baselines reliable?* Most of the recently introduced MU methods are compared only with three baselines: Fine-tuning, Gradient Ascent, and Successive Random Labels, and not across recently introduced approaches. We show that with proper hyperparameter selection FT is a reliable baseline. In contrast, GA consistently performs poorly and should be replaced by the more recent Negative Gradient Plus (NG+) [37] that simultaneously reduces the performance in the forget set while maintaining the performance on the rest of the data points.

*Q2. How reliable are MU methods across datasets, and, models, and initializtions?* Our findings show that, unlike the majority of recent MU methods, Masked-Small-Gradients is consistently among the best performing methods across various metrics. In contrast, methods such as SRL do not consistently outperform others across different datasets.

*Q3. Among existing methods, which are the most reliable and accurate?.* Among 18 methods, our evaluation shows that certain methods, such as Masked-Small-Gradients (MSG), Convolution-Transpose (CT), and Fine-tuning (FT), exhibit desirable properties. Specifically, MSG and CT show resilience against U-LiRA, a strong per-sample membership inference attack. Furthermore, all three methods show consistency across datasets, and these methods could serve as reliable baselines for future studies.

We publish the source code used for reproducing the experiments conducted in this paper at *[released upon publishing this paper]*.

## 2   Background of Machine Unlearning

**Setting.** Starting with a **training set** $\mathcal{D} = \{(\mathbf{x}_i, y_i)\}_{i=1}^{N}$ and a trained model $f_O$, referred to as the **original model**, the objective of a MU method $U$ is to *remove* the influence of a particular subset of training set $\mathcal{D}_F = \{(\mathbf{x}_i, y_i)\}_{i=1}^{K} \subset \mathcal{D}$ referred to as the **forget set** where $K \ll N$. The rest of training set $\mathcal{D}_R = \mathcal{D} \backslash \mathcal{D}_F$ is called the **retain set**. The forget set and the retain set are distinct and complementary subsets of the training set. The outcome of MU is an **unlearned** model $f_U$, the aim for which is to *perform* on par with a model *retrained from scratch* on $\mathcal{D}_R$; this latter model $f_R$ is referred to as the **retrained** model. We denote the weights of the original model and the retrained model as $\theta_O$ and $\theta_R$, respectively. For evaluations, we consider two held-out sets: the **validation set** $\mathcal{D}_V$ and **test set** $\mathcal{D}_T$, both drawn from the same distribution as $\mathcal{D}$. We consider the accuracy of the retrain model as the optimal accuracy. One critical assumption we make is that the MU method has access to $\theta_O, \mathcal{D}_R, \mathcal{D}_F$, and $\mathcal{D}_V$.

**(1) Unlearning Evaluation.** To evaluate the success of unlearning, one approach is to check whether data points in $\mathcal{D}_F$ still influence the predictions made by the unlearned model [13, 37, 31]. This is commonly done via *influence functions* [35, 8, 29], membership inference attacks (MIA) [44] MIA has become one of the most common approaches for evaluating unlearning algorithms. It aims to determine whether specific data points were part of the original training dataset based on the unlearned model. We compute the population-based U-MIA, denoted with $\text{MIA}(\mathcal{D}, f_U)$ evaluated on data $\mathcal{D}$[1]. Using this we define the **discernibility** metric as $\text{Disc}(\mathcal{D}_V, f_U) = |2 \times \text{MIA}(\mathcal{D}, f_U) - 1| \in [0, 1]$, and similarly, **indiscernibility** is given by $\text{Indisc}(\mathcal{D}, f_U) = 1 - \text{Disc}(\mathcal{D}, f_U) \in [0, 1]$. We set $\mathcal{D}$ to either the test set $\mathcal{D}_T$ or validation set $\mathcal{D}_V$. The indiscernibility equals 1 when the accuracy of the MIA is not better than random guessing. A more recent MIA variation is the unlearning likelihood ratio attack (U-LiRA) [37]. As highlighted by [31], U-LiRA is a more robust evaluation approach for approximate MU. Nonetheless, U-LiRA is much more computationally demanding than U-MIA. We evaluate the methods that defeat the weaker, less expensive U-MIA attack, additionally against the U-LiRA attack.

**(2) Accuracy.** The classification accuracy of unlearned model on the retain set should be as close as possible to that of the original model. First, we consider three metrics derived directly from the model's classification accuracy on different sets: **retain accuracy** (RA), **forget accuracy** (FA), and **test accuracy** (TA). RA is defined as

$$\text{RA}(\mathcal{D}_R, f_U) = \frac{1}{|\mathcal{D}_R|} \sum_{(\mathbf{x}_i, y_i) \in \mathcal{D}_R} \mathbf{1}_{y_i = f_U(\mathbf{x}_i)} \in [0, 1]. \tag{1}$$

The metrics for FA and TA can be derived by replacing $\mathcal{D}_R$ with $\mathcal{D}_F$ and $\mathcal{D}_T$, respectively. Second, while RA, FA, and TA give us insight into the overall accuracy of the unlearned model, they do not capture how well it performs compared to a retrained model $f_R$. Considering the $f_R$ model as a gold standard, we derive three more metrics: **retain retention** (RR), **forget retention** (FR), **test retention** (TR), where RR is given by,

$$\text{RR}(f_U, f_R) = \frac{\text{RA}(\mathcal{D}_R, f_U)}{\text{RA}(\mathcal{D}_R, f_R)} \in [0, +\infty), \tag{2}$$

and formulas for FR and TR can be derived using FA and TA, respectively. An unlearned model with a score of 1 indicates that its accuracy perfectly matches the accuracy of the reference retrain model. A score below 1 indicates that the model under-performs and a score above 1 indicates that the model over-performs. We further define the **retention deviation** (RetDev) as:

$$\text{RetDev} = |RR(f_U, f_R) - 1| + |FR(f_U, f_R) - 1| + |TR(f_U, f_R) - 1| \in [0, +\infty), \tag{3}$$

which provides information on the cumulative divergence of the unlearned model in terms of retention score. The closer to 0, the better as 0 indicates that the model perfectly matches the performance of the retrained model.

**(3) Efficiency.** run-tiee efficiency (RTE) of a MU method should ideally be lower than the naive approach of just retraining the model from scratch on the retain set. As the high computational cost of the retraining algorithm motivated the development of approximate MU methods, we evaluate how much faster each MU method is compared to the retraining algorithm. We define RTE of $U$ as the number of seconds it takes to complete, denoted as $\text{RT}(U)$ (we use the same machine and resources for all the experiments). To indicate the relative speedup compared to the retrained model, we define RTE of an unlearn method $U$ as:

$$\frac{\text{RT}(U_R)}{\text{RT}(U)} \in [0, +\infty), \tag{4}$$

where $U_R$ denotes the retrain method. The RTE of retraining from scratch would thus be 1; any method with an RTE less than 1 is slower than retraining from scratch, and vice versa.

---

[1]The methodology for this follows the classic MIA: we compute the losses on $\mathcal{D}_F$ and $\mathcal{D}_V$, we shuffle and trim them so that they are of equal size. We then train logistic regression models in a 10-fold cross validation, and compute the average accuracy across the folds.

# 3 Machine Unlearning Methods

We briefly discuss the main unlearning methods considered.

## 3.1 Classical Baselines

**FineTune (FT)** finetunes the original model $f_O$ on only the retain set $\mathcal{D}_R$ for several epochs.

**Successive Random Labels (SRL)** the model is trained on both the forget set $\mathcal{D}_F$ and $\mathcal{D}_R$ where the labels of $\mathcal{D}_F$ are randomly assigned at each epoch.

**Gradient Ascent (GA)** trains the model using gradient *ascent* steps on the $\mathcal{D}_F$.

## 3.2 State-of-the-art MU methods

Expanding upon the classical baselines, we additionally evaluate 15 recent MU methods. We first discuss the seven top-performing methods from the Machine Unlearning Competition 2023 on Kaggle.

**Forget-Contrast-Strengthen (FCS)** [1] minimizes the Kullback-Leibler Divergence (KLD) between the model's output on $\mathcal{D}_F$ and a uniform distribution over the output classes, then alternatively optimizes a contrastive loss between the model's outputs on $\mathcal{D}_R$ and $\mathcal{D}_F$, and minimizes the cross-entropy loss on $\mathcal{D}_R$.

**Masked-Small-Gradients (MSG)**[2] accumulates gradients via gradient *descent* on the $\mathcal{D}_R$ and gradient *ascent* on the $\mathcal{D}_F$, then reinitialize weights with the smallest absolute gradients while dampening subsequent weights updates on the $\mathcal{D}_R$ for the other weights.

**Confuse-Finetune-Weaken (CFW)**[3] injects noise into the convolutional layers and then trains the model using a class-weighted cross-entropy on $\mathcal{D}_R$, then injects noise again toward the final epochs.

**Prune-Reinitialize-Match-Quantize (PRMQ)** [4] first prunes the model via L1 pruning, reinitializes parts of the model, optimises it using a combination of cross-entropy and a mean-squared-error on the entropy between the outputs of $f_O$ and $f_U$ on $\mathcal{D}_R$ and finally converts $f_U$'s weights to half-precision floats.

**Convolution-Transpose** [5] simply transposes the weights in the convolutional layers and trains on $\mathcal{D}_R$.

**Knowledge-Distillation-Entropy (KDE)** [6] uses a teacher-student setup. Both student and teacher start as copies of the original model, then the student's first and last layers are re-initialised. The student $f_U$ minimizes its Kullback-Leibler Divergence (KLD) with the $f_O$ over $\mathcal{D}_V$, then minimizes a combination of losses: a soft cross-entropy loss between $f_U$ and $f_O$, a cross-entropy loss on outputs of $\mathcal{D}_R$ from $f_U$, and the KLD between $f_U$ and $f_O$ on $\mathcal{D}_R$.

**Repeated-Noise-Injection (RNI)** [7] first reinitialises the final layer of the model, then repeatedly injects noise in different layer of the model while training on the $\mathcal{D}_R$.

We further consider eight state-of-the-art methods introduced in the literature.

**Fisher Forgetting (FF)** [25, 20] adds noise to $f_O$ with zero mean and covariance determined by the 4th root of Fisher Information matrix with respect to $\theta_O$ on $\mathcal{D}_R$.

**Influence Unlearning (IU)** [33, 47, 34] uses Influence Functions[18] to determine the change in $\theta_O$ if a training point is removed from the training loss. IU estimates the change in model parameters from $\theta_O$ to the model trained without a given data point. We use the first-order WoodFisher-based approximation from [34].

**Catastrophic Forgetting - K (CF-K)** [24] freezes the first layers then trains the last $k$ layers of the model on $\mathcal{D}_R$.

**Exact Unlearning - K (EU-K)** [24] freezes the first layers then restores the weights of the last $k$ layers to their initialization state. We randomly reinitialize the weights instead, so that the method no longer requires knowledge about the training process of $f_O$.

**SCRUB** [37] leverages a student-teacher setup where the model is optimised for three objectives: matching the teacher's output distribution on $\mathcal{D}_R$, correctly predicting the $\mathcal{D}_R$ set and ensuring the output distributions of the teacher and student diverge on the $\mathcal{D}_F$

**Saliency Unlearning (SaLUN)** [20] determines via gradient *ascent* which weights of $\theta_O$ are the most relevant to $\mathcal{D}_F$, then trains the model simultaneously on $\mathcal{D}_R$ and $\mathcal{D}_F$ with random labels on $\mathcal{D}_F$, while dampening the gradient propagation based on the selected weights.

**Negative Gradient Plus (NG+)** [37] is an extension of the Gradient Ascent approach where additionally a gradient descent step is taken over the $\mathcal{D}_R$.

**Bad Teacher (BT)** [16] uses a teacher-student approach with two teachers: the original model, and a randomly initialized model - the bad teacher-, the student starts as copy of $f_U$ then learns to mimic the $f_O$ on $\mathcal{D}_R$ and the bad teacher on the $\mathcal{D}_F$.

# 4   Experimental Evaluation

**Experiments.** We evaluate the 18 recent MU methods as described in Section 3 across 5 benchmark datasets: MNIST [38], FashionMNIST [49], CIFAR-10 [36], CIFAR-100 [36], and UTK-Face [53]. These datasets vary in difficulty, number of classes, instances per class, and image sizes. We consider two model architectures: a TinyViT and a ResNet18 model. Hence, in total we evaluated nine different combinations of models and architectures: ResNet18 and TinyViT on MNIST, Fashion-MNIST, CIFAR-10, CIFAR-100, and ResNet18 on UTKFace. More information on the data sets, hyperparameters, and data augmentations used to train the original and retrained models is provided in the appendix B. *We construct the forget set by sampling 10% of $\mathcal{D}$.*

The performance of the MU methods can change across datasets, model configurations, and model initializations; a reliable MU method remains consistent across these changes. For each method, dataset and model combination, we unlearn from Original models initialized using 10 different seeds and consider the average performance across seeds.

A further observation is that prior research tends to compare MU methods with default hyperparameters, potentially leading to a less competitive performance of the method. To ensure that we get the best performance out of each method, we perform three hyperparameter sweeps to find the best set of hyperparameters for each method. To ensure a fair comparison, we use same number of searches for each method. Each hyper-parameter sweep uses 100 trials to minimize four loss functions: Retain Loss ($\mathcal{L}_{\text{Retain}}$), Forget Loss ($\mathcal{L}_{\text{Forget}}$), Val Loss ($\mathcal{L}_{Val}$), and Val MIA ($\mathcal{L}_{\text{Val-MIA}}$) given by

$$\mathcal{L}_{Retain} = \alpha \times |\text{RA}(f_U) - \text{RA}(f_R)|, \ \mathcal{L}_{Forget} = \beta \times |\text{FA}(f_U) - \text{FA}(f_R)|, \tag{5}$$

$$\mathcal{L}_{Val} = \gamma \times |\text{VA}(f_U) - \text{VA}(f_R)|, \ \mathcal{L}_{\text{Val-MIA}} = \eta \times \text{Disc}(\mathcal{D}_V, f_U), \tag{6}$$

where the $\mathcal{L}_{\text{Retain}}$ captures the divergence in accuracy between the retrained and unlearned model over the $\mathcal{D}_R$, $\mathcal{L}_{\text{Forget}}$ and $\mathcal{L}_{Val}$ capture the divergence over $\mathcal{D}_F, \mathcal{D}_V$ respectively and $\mathcal{L}_{\text{Val-MIA}}$ captures whether the loss distributions over $\mathcal{D}_F$ and $\mathcal{D}_V$ are distinguishable from one another via the discernibility score defined in Section 2. We set $\alpha = \beta = \gamma = \frac{1}{3}$ and $\eta = 1$ as we found these values to balance the importance of importance retention and the resilience to Membership Inference Attacks. Per unlearn method, we use the hyperparameter configuration that minimises the four loss terms when evaluating the method. Thus, for each unlearning method, we first unlearn 300 models to do the hyper-parameter sweep, then unlearn 10 models with the best set of hyper-parameters, leading to $5,580$ per dataset for a given architecture, leading to a total of $50,220$ for the 9 dataset / model combinations.

**Ranking.** A challenge in the comparison of MU method performance comes from the potential proximity of the evaluation metrics. As a simple example, suppose we have four methods $U_1, ..., U_4$

Table 1: Ranking by performance on Retention Deviation and Indiscernibility across datasets and architectures. We count the number of times each method appears in the Best Performers group (G1), Average performance group (G2) and Worst performers group (G3) (see §4). The final rank is computed based on the number of times the method appears in G1—with occurrences in G2 and G3 used to break ties if needed. If a method does not produce any usable models, it is assigned to a Failed group (F). Three methods appear in the top 3 for both performance measures: MSG (1st and 1st), CT (3rd and 1st) and KDE (3rd and 2nd).

| | | Retention Deviation | | | | | | Indiscernibility | | | |
|---|---|---|---|---|---|---|---|---|---|---|---|
| Rank | Method | G1 | G2 | G3 | F | Rank | Method | G1 | G2 | G3 | F |
| 1 | FT | 8 | 1 | 0 | 0 | 1 | CT | 9 | 0 | 0 | 0 |
| 1 | MSG | 8 | 1 | 0 | 0 | 1 | MSG | 9 | 0 | 0 | 0 |
| 2 | PRMQ | 7 | 2 | 0 | 0 | 2 | CFW | 7 | 2 | 0 | 0 |
| 3 | CT | 7 | 1 | 1 | 0 | 2 | RNI | 7 | 2 | 0 | 0 |
| 3 | KDE | 7 | 1 | 1 | 0 | 2 | KDE | 7 | 2 | 0 | 0 |
| 3 | CFW | 7 | 1 | 1 | 0 | 3 | FT | 6 | 3 | 0 | 0 |
| 4 | FCS | 6 | 3 | 0 | 0 | 3 | PRMQ | 6 | 3 | 0 | 0 |
| 4 | SalUN | 6 | 3 | 0 | 0 | 3 | SalUN | 6 | 3 | 0 | 0 |
| 5 | NG+ | 5 | 4 | 0 | 0 | 4 | SRL | 6 | 2 | 1 | 0 |
| 5 | SRL | 5 | 4 | 0 | 0 | 5 | NG+ | 5 | 4 | 0 | 0 |
| 6 | SCRUB | 4 | 3 | 1 | 1 | 5 | FCS | 5 | 4 | 0 | 0 |
| 7 | BT | 2 | 7 | 0 | 0 | 6 | SCRUB | 5 | 3 | 0 | 1 |
| 7 | RNI | 2 | 7 | 0 | 0 | 7 | BT | 4 | 5 | 0 | 0 |
| 8 | CF-k | 2 | 3 | 2 | 2 | 8 | CF-k | 1 | 2 | 4 | 2 |
| 9 | IU | 1 | 0 | 2 | 6 | 9 | EU-k | 1 | 2 | 2 | 4 |
| 10 | EU-k | 0 | 5 | 0 | 4 | 10 | GA | 0 | 4 | 4 | 1 |
| 11 | GA | 0 | 1 | 7 | 1 | 11 | IU | 0 | 0 | 3 | 6 |
| 12 | FF | 0 | 0 | 0 | 9 | 12 | FF | 0 | 0 | 0 | 9 |

with accuracies: 98%, 99%, 50%, 1%, respectively; if we simply rank the methods, the rank itself would not be representative of the fact that e.g. $U_1$ and $U_2$ are much above $U_3$ and $U_4$. In order to enable distinctions based on proximities, we use Agglomerative Clustering and define cut-off points such that we obtain three clusters: (1) Best performers (G1), (2) Average performers (G2), and (3) Worst performers (G3). If a method does not produce 10 usable models, one per original model, it is assigned to a Failed group (F). For each method, we count the number of times it appears in each of the three groups (with nine being the maximum). To obtain a final ranking of the methods, we first rank the methods using the number of times it appears in the Best Performers group (G1); if ties occur, we use the Average Performers (G2) group to break them. If ties persist, the Worst Performers (G3) group serves as the final tie-breaker. This method ensures a clear and fair ranking by considering each performance group in order of importance.

## 5 Main Results

Table 1 presents the main results of our evaluations on MU methods based on Retention Deviation and Indiscernibility. The results for the run-time efficiency are shown in Table 2.

**On the reliability of baselines.** The commonly used baseline FT trains the original model only on the retain set for several epochs to enable the model to forget information about the forget set. In our evaluation, FT performs best based on the Retention Deviation, and is ranked third based on Indiscernibility. The latter observation may come from the fact that FT does not explicitly unlearn the forget set or perturb the model parameters. Based on these results we conclude it is a reasonable baseline to evaluate against. We however remark that since the mechanism underlying FT (training on the Retain set to maintain performance) is common to many other methods, these methods may inherit its susceptibility to MIA. Another common baseline, GA which performs gradient ascent on

Table 2: Run Time Efficiency on ResNet for the top performing methods. CT is the fastest on average, MSG runs up to 5x faster than naive retraining.

| Unlearner | CIFAR-10 | CIFAR-100 | MNIST | FashionMNIST | UTKFace | Average |
|---|---|---|---|---|---|---|
| MSG | 6.80 | 4.49 | 4.29 | 3.32 | 7.57 | 5.29 |
| CFW | 4.67 | 6.17 | 4.29 | 4.90 | 5.54 | 5.11 |
| PRMQ | 4.93 | 4.34 | 3.77 | 3.77 | 5.88 | 4.54 |
| CT | 17.49 | 11.82 | 5.83 | 4.47 | 13.34 | 10.59 |
| KDE | 6.33 | 3.98 | 3.27 | 3.22 | 8.19 | 5.00 |
| FT | 8.15 | 5.16 | 5.07 | 4.29 | 5.78 | 5.69 |

the forget set, performs poorly across both metrics. Its more recent variation NG+, which uses an additional retain set correction, ranks fifth for both metrics, making it a more suitable baseline.

**On the reliability of newly proposed unlearning methods.** MSG obtains a first rank in both Performance Retention Deviation and Indiscernibility. Its unique approach identifies the parameters in the Convolutional layers that most contribute to the information to be forgotten. This strategy, differing from FT, allows MSG to retain performance from the Retain set while modifying the weights that are more relevant to the Forget set. PRMQ ranks second in performance retention, making it one of the top performers. However, it suffers from the same lower performance in Indiscernibility as FT. PRMQ however does not leverage the Forget set; instead, it performs a form of knowledge distillation by attempting to reproduce the results of the original models on the Retain set. Additionally, during the pruning phase, it reinitializes the weights for the MLP and Convolutional layers. CT ranks third in Performance Retention and first in Indiscernibility. It is interesting to note that CT and MSG are both consistently among the top performers in Indiscernibility, where FT performs poorly.

**On the robustness across architectures.** Critical for a good MU method is its ability to generalise across various DNN architectures. We conducted experiments with both ResNet18 and TinyViT. Methods such as CT, despite being tailored to Convolutional Neural Networks (CNN) models, still perform competitively when applied to Vision Transformers. Methods such as CT and MSG, while proposed for CNN layers, work well on Vision Transformer as one can leverage the 2D Convolutions used in Positional Encoders. We provide additional details on the ranking based on architectures in Appendix F.

**On the speed of the unlearn methods.** MSG, the best MU candidate runs 7.6x faster than retraining from scratch on UTKFace and 3.3x on FashionMNIST. On average MSG runs 5.3x faster than retraining. The fastest method is CT which achieves a speedup of 17.5x compared to Retraining. This speedup stems from its simple approach: transpose the weights of the convolutional layers.

**On the performance when evaluated against a stronger MIA.** From the results above, we note that there exist MU methods that perform fast and reliably across datasets, architectures, and initializations. Specifically, CT, FT, MSG seem strong candidates for reliable MU methods. However, as has also been highlighted in prior work [46], MIA has been debated as a strong metric, as its ability to assess MU is hampered by its own ability to infer data membership. In line with this, recent works have introduced more powerful variations on MIA [51] with [37] proposing the stronger U-LiRA attack for MU. For the best performers in Table 1, we apply the attack setup from [31] where we generate a total of 640 models (with varying train, retain and forget sets) and for each unlearn method perform a hyperparameter sweep to find the best configuration (for details see Appendix D). We determine for each data point its U-LiRA Inference Accuracy and report for each method its average and standard deviation (see Figure 1). From this, we conclude that MSG as well as CT resist U-LiRA attacks. Both MSG and CT thus not only rank first in terms of Performance Retention and Indiscernibility based on U-MIA, but also are robust against a stronger variation of MIA.

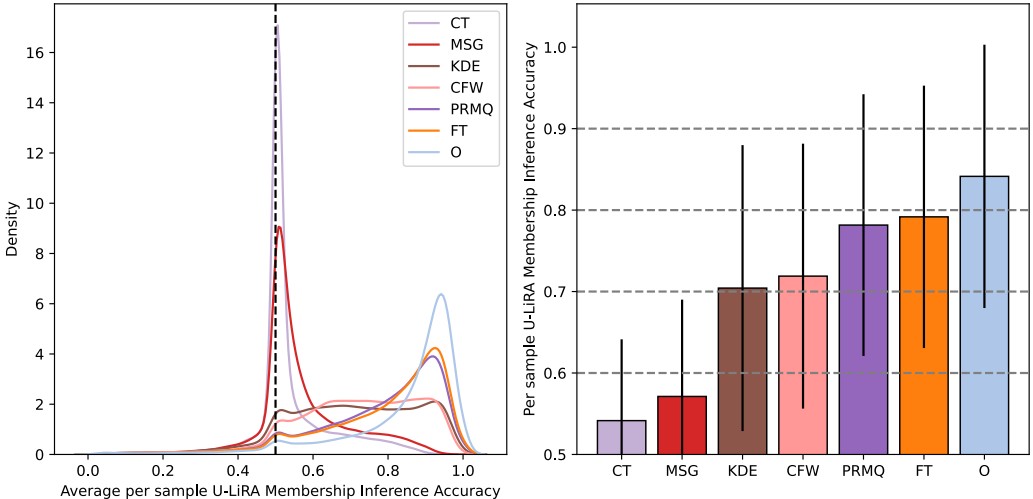

Figure 1: U-LiRA on CIFAR-10 on ResNet models. Both CT and MSG, which ranked first against U-MIA, showed great resilience against the U-LiRA attack.

## 6 Discussion and Conclusion

The increasing focus on data privacy and trustworthiness of machine learning models underscores the need for robust and practical methods to unlearn and remove the influence of specific data from trained models. Due to the growing size of models, we require methods that avoid the computationally costly retraining from scratch. In this work we performed a comprehensive comparison of approximate unlearning methods across various models and datasets aimed to address this critical issue.

We experimentally compared 18 methods across different datasets and architectures, focusing on assessing the method's ability to maintain privacy and accuracy while being computationally efficient and reliable across datasets, architectures and random seeds. Our findings indicate that Masked-Small-Gradients, which accumulates gradients via gradient descent on the data to remember and gradient ascent on the data to forget to determine which weights to update, consistently outperforms for all metrics across the studied datasets, architectures, and initialization seeds. Similarly, Convolution Transpose, which leverages the simple transposition in convolutional layers, performed strongly.

Both CT and MSG were resistant against both a population-based Membership Inference Attack (MIA) and a stronger, per-sample attack (U-LiRA). However, a core challenge of approximate unlearning is that these methods will only be as strong as the attacks against which they are tested. As stronger and more complex attacks emerge, some approximate unlearning methods might no longer be as efficient as initially expected. This highlights the need for continuous evaluation and adaptation of unlearning methods to maintain their effectiveness. We also conducted experiments based on L2 distances, but found that no method consistently got close to the reference models' weights, we provide further information in Appendix G.

**Limitations.** Due to computational costs, we limited our analysis to Tiny Vision Transformers and ResNet; a further investigation of other architectures could provide useful insights. We did not investigate different amounts of unlearning samples, which some methods are known to be sensitive to [37]. We did not consider repeated deletion, instead we assume that there is a single forget set and that the unlearning process happens once, as is common in the literature, nonetheless, in practical applications one might need to unlearn different smaller forget sets over time and some unlearning methods might not work as well under such scenario. We finally remark once again on the difficulty of evaluation for approximate unlearning [31]: while these methods provide significant gains in efficiency, novel attacks might highlight yet unknown weaknesses of the unlearning processes.

**Future work.** First, we put our focus on natural image data, however, machine unlearning is relevant to other data types such as medical images or other modalities such as time series, audio and speech, or language data. Second, we focus on the classification task, however, other learning tasks would greatly benefit from machine unlearning too. For instance removing concepts from generative models for images [20] or poisoned data in language models [31]. Third, this work focuses on empirically benchmarking approximate machine unlearning methods. We do not provide a theoretical analysis of these methods or a rigorous comparison with exact unlearning algorithms.

**Impact statement.** This paper aims to highlight the importance of effectively assessing approximate machine unlearning methods. Our goal is to stress the need for evaluating new unlearning methods against more reliable baselines and experimental setups. Additionally, it is crucial to assess the consistency of a new unlearning method across various datasets and model architectures. Without such a thorough evaluations, proposed unlearning methods may provide a false sense of privacy and safety, ultimately limiting their effectiveness for data regulation.

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
