# Appendix

The appendix is structured as follows:

- **Related Works (Section A)** provides an overview of the previous works related to benchmarking MU.

- **Datasets (Section B)** contains information about the datasets, data augmentations, and hyper-parameters used to train the Original and Retrained models.

- **Neural Network Architectures (Section C)** provides information about ResNet18 and TinyViT used throughout our benchmark.

- **Privacy Evaluation (Section D)** gives the descriptions of the U-MIA and U-LiRA evaluation metrics.

- **Per Dataset Results (Section E)** shows some experimental results in terms of accuracy, retention, privacy metrics, and runtime efficiency per dataset for the 9 combinations of datasets and DNN architectures considered in our work.

- **Per Architectures Ranking (Section F)** provides Performance Retention Deviation and Indiscernibility Rankings separated for ResNet18 and TinyViT.

- **L2 Distances between Model Weights (Section G)** shows L2 distances computed between the Unlearned, Original, and Retrained models.

- **Requirements (Section H)** which describes the compute resources.

# A  Related Works

## A.1  Machine Unlearning

Machine Unlearning is often first associated with the work from Cao et al. [11], followed by Bourtoule et al. [9], which proposes SISA (Sharded, Isolated, Sliced, Aggregated) as an exact unlearning method. For a recent overview of MU, we refer to the survey from Xu et al. [50], which provides a taxonomy of common unlearning methods. Furthermore, Zhang et al. [52] review MU through privacy-preserving and security lenses. The authors cover the Confidentiality, Integrity, and Availability security triad and the need for Data Lineage, which relates to following the movement of data in a machine learning pipeline and understand from where it originates, where it is stored, and how it percolates in the system through transformation. Some might have information on common MU verification methods, privacy evaluation metrics, and datasets, we defer to the work of Nguyen et al. [39], and Shaik et al. [43].

In the following, we focus on the MU taxonomy from Xu et al. [50], which considers Data Reorganization and Model Manipulation:

**(1) Data Reorganization** methods focus on directly modifying the data to perform unlearning. It is divided into Data Obfuscation, Data Pruning, and Data Replacement.

*Data Obfuscation* refers to modifying the dataset to obscure the influence of the data to be unlearned: random relabeling and retraining [28], SRL (Successive Random Labels), and Saliency Unlearning (SalUN).

*Data Pruning* usually relies on dividing the dataset into multiple sub-datasets and training sub-models on these subsets. This is the category to which SISA [9] relates. Our work does not consider methods associated with this setting as they assume the training process.

*Data Replacement* attempts to unlearn by replacing the original dataset with transformed data that simplifies unlearning specific samples. For instance, Cao et al. [11], replace the training data with summations of efficiently computable transformations. Like data pruning, these methods tend to make strong assumptions about the training process.

**(2) Model Manipulation** methods directly adjust the model parameters to remove the influence of specific data points. Model manipulation is divided into Model Shifting, Model Replacement, and Model Pruning.

*Model Shifting* directly updates the model parameters to offset the influence of the unlearned samples, such as using a single step of Newton's method on model parameters [30] or decremental updates [40], in our benchmark Fisher Forgetting (FF), Influence Unlearning (IU), and SalUN would represent these approaches.

*Model Replacement* uses pre-calculated parameters that do not reflect the data to forget to replace parts of the trained model. For instance, when using decision trees, one can replace nodes affected by the forget set by pre-calculated node [41]. These methods often make strong assumptions about the training process and the overall model.

*Model Pruning* prunes specific parameters from the trained models to remove the influence of certain samples [34] or Prune-Reinitialize-Match-Quantize (PRMQ) [4] which prunes the model via L1 pruning, reinitializes parts of the model then train the model on $\mathcal{D}_R$.

## A.2  Machine Unlearning for Deep Neural Networks

Initially, MU research primarily focused on linear models such as linear regression and logistic models. Such models allow for the design of methods that assume the convexity of the loss function, rendering them less practical for DNN-based approaches. Since DNNs are able to memorize parts of their training data, they are particularly relevant targets for MU, even more so when they have been trained on large amounts of potentially personal data. For Deep Learning models, unlearning raises

additional challenges: 1) the non-convexity of the loss function of Deep Neural Networks [15], 2) the size of the models inducing large computational costs, 3) the randomness coming from the model's training process, such as the initialization seed, randomness in the mini-batch generation process, and 4) the fact that any model update impacts subsequent versions of the models, namely the weights at epoch $n + 1$ directly depend on the weights at update $n$.

When considering MU for DNN, Xu et al. [50] notes that a standard scheme DNN is to focus only on the final layer, as it is expected for this layer to be the most relevant for the downstream task, and stems from the early works MU. Nonetheless, Goel et al. [24] showed that simply modifying the final layer is often insufficient to remove information related to $\mathcal{D}_f$. However, other approaches, such as those from Golatkar et al. [25, 26, 27], attempt to unlearn the full model via methods derived from Information Theory. For instance, weight scrubbing on trained models can be done by approximating the Fisher information matrix.

### A.3 Post-Hoc Machine Unlearning

While proactively designing deep-learning pipelines with built-in unlearning methods such as SISA can greatly simplify the unlearning process, many contemporary services relying on DNNs were not deployed with unlearning in mind. This motivates searching for methods that can unlearn from already trained models without making assumptions about the training process.

Thus, we focus on *post-hoc MU*, a scenario where we assume that the unlearning method is agnostic to the original training process of the model. Under such a scenario, differences exist in terms of data availability at unlearning time. For instance, whether one has access to the original training data $\mathcal{D}$, the retain set $\mathcal{D}_R$, the forget set $\mathcal{D}_F$, or even some external set such as the validation set $\mathcal{D}_V$. Therefore, careful consideration should be given to the data requirement associated with an unlearning method. Indeed, some might require having access to both $\mathcal{D}_R$ and $\mathcal{D}_F$ at the unlearning time, while others assume that $\mathcal{D}_R$ is no longer available [17] making them more practical in real-world scenarios. Throughout our benchmark, we make the same assumption as the NeurIPS2023 Unlearning Challenge [45], where the unlearning methods had access to $f_O, \mathcal{D}_R, \mathcal{D}_F, \mathcal{D}_V$

### A.4 Machine Unlearning and Differential Privacy

We based our Unlearning definition on Sekhari et al. [42] and refer to their work on the distinction between Differential Privacy and the objective of Machine Unlearning. Differential privacy, in a high-level picture, is a method for publicly sharing aggregated information about a population by describing the patterns discovered among the groups within the dataset while withholding specific information about individual data points. A randomized algorithm $\mathcal{A}$ is $(\varepsilon, \delta)$-differentially private if for all datasets $D_1$ and $D_2$ that differ on a single data point, and all $S \subseteq \text{Range}(\mathcal{A})$,

$$\Pr[\mathcal{A}(D_1) \in S] \leq e^{\varepsilon} \cdot \Pr[\mathcal{A}(D_2) \in S] + \delta. \tag{7}$$

In this definition, $\varepsilon$ (epsilon) is a non-negative parameter that measures the privacy loss, with smaller values indicating stronger privacy. The parameter $\delta$ represents the probability of breaking differential privacy, ideally close to or equal to 0.

Despite enabling provable error guarantees for Unlearning methods, Differential Privacy requires strong model and algorithmic assumptions, making MU, derived from it, potentially less effective against practical adversaries [34].

## B   Datasets

**CIFAR-10**   CIFAR-10 is a widely used dataset in computer vision and machine learning. It comprises 60,000 32x32 color images in 10 different classes, with 6,000 images per class. The dataset is divided into 50,000 training images and 10,000 testing images. CIFAR-10 represents a diverse range of everyday objects, such as airplanes, automobiles, birds, and cats, making it a challenging

task for image classification. The simplicity of the images combined with the variety of categories makes CIFAR-10 a suitable dataset to test the efficacy of machine unlearning algorithms in effectively unlearning information without compromising the model's performance on the remaining data.

*Data Augmentations:* random cropping to 32x32 with 4-pixel padding, 50% random horizontal flipping, and per-channel normalization with a mean of $[0.4919, 0.4822, 0.4465]$ and standard deviation of $[0.2023, 0.1994, 0.2010]$. At test time, we resize to 32x32 and normalize.

**CIFAR-100**  CIFAR-100 is a more complex extension of CIFAR-10, containing 100 classes with 600 images per class, split into 500 training images and 100 testing images per class. Each class is labeled with a "fine" label and grouped into 20 "coarse" labels, adding another layer of classification difficulty. The increased number of classes and finer granularity make CIFAR-100 an intriguing dataset for machine unlearning benchmarks. It poses a more significant challenge for models to forget specific classes or groups while retaining knowledge of others, thus testing the unlearning algorithms' precision and effectiveness in handling more granular and complex datasets.

*Data Augmentations:* random cropping to 32x32 with 4-pixel padding, 50% random horizontal flipping, and per-channel normalization with a mean of $[0.5071, 0.4865, 0.4409]$ and standard deviation of $[0.2673, 0.2564, 0.2762]$. At test time, we resize to 32x32 and normalize.

**MNIST**  The MNIST dataset is a well-known benchmark in handwritten digit recognition. It comprises 70,000 grayscale images of handwritten digits (0-9), 60,000 used for training, and 10,000 for testing. Each image is 28x28 pixels in size. We consider MNIST due to its simplicity and extensive research and development history. The simplicity of MNIST allows researchers to focus on the fundamental aspects of unlearning techniques without the additional complexity introduced by color or high resolution, providing a clear assessment of the effectiveness of unlearning algorithms in a controlled setting.

*Data Augmentations:* conversion to 3 channels, resizing to 32x32 such that both ResNet18 and TinyViT use the same input resolution, 50% random horizontal flipping, and per-channel normalization with a mean of $[0.1307, 0.1307, 0.1307]$ and standard deviation of $[0.3081, 0.3081, 0.3081]$. We convert to 3 channels at test time, resize to 32x32, and normalize.

**Fashion MNIST**  Fashion MNIST is a more challenging replacement for MNIST. It contains 70,000 grayscale images of fashion items in 10 categories: shirts, trousers, and sneakers. Like MNIST, each image is 28x28 pixels, but the increased complexity and variability of clothing items make it a more challenging classification task. Fashion MNIST provides a more realistic and intricate dataset than MNIST, testing the unlearning algorithms' ability to handle real-world-like variability and ensuring that they can effectively remove learned information while maintaining performance on a moderately complex dataset.

*Data Augmentations:* conversion to 3 channels, resizing to 32x32, 50% random horizontal flipping, and per-channel normalization with a mean of $[0.2860, 0.2860, 0.2860]$ and standard deviation of $[0.3560, 0.3560, 0.3560]$. We convert to 3 channels at test time, resize to 32x32, and normalize.

**UTKFace**  UTKFace is a large-scale face dataset containing over 20,000 images of faces with annotations of age, gender, and ethnicity. The images vary in size and cover a wide range of ages, from 0 to 116. UTKFace is particularly interesting due to the sensitive nature of the data and the need for privacy-preserving techniques.

*Data Augmentations:* resizing to 224x224, and per-channel normalization with a mean of $[0.485, 0.456, 0.406]$ and standard deviation of $[0.229, 0.224, 0.225]$. We apply the same transformation at test time.

For each dataset, the Original and Retrained models are trained using the same hyper-parameters (provided in Table 3)

Table 3: Summary of the number epochs, learning rate, and batch size for each dataset and model used to train the Original and Retrained models.

| Dataset | Model | Epochs | Learning Rate | Batch Size |
|---|---|---|---|---|
| FashionMNIST | ResNet18 | 50 | 0.1 | 256 |
| | TinyViT | 50 | 0.1 | 256 |
| MNIST | ResNet18 | 50 | 0.1 | 256 |
| | TinyViT | 50 | 0.1 | 256 |
| CIFAR-10 | ResNet18 | 182 | 0.1 | 256 |
| | (LiRA) ResNet18 | 91 | 0.1 | 256 |
| | TinyViT | 182 | 0.1 | 256 |
| CIFAR-100 | ResNet18 | 182 | 0.1 | 256 |
| | TinyViT | 182 | 0.1 | 256 |
| UTKFace | ResNet18 | 50 | 0.1 | 128 |
| | TinyViT | 50 | 0.1 | 128 |

# C  Neural Network Architectures

We consider two families, ResNet (Residual Network) [32] and ViT (Vision Transformer) [19], which are prominent architectures in computer vision. We consider ResNet18 and a TinyViT[48] with approximately 11M learnable parameters for a fair comparison between two fundamentally different architectures. This provides insights into how architectural differences impact the unlearning process and helps understand the trade-offs between convolutional and transformer-based models regarding reliability and computational efficiency.

**ResNet: ResNet18**  Introduced by He et al. [32], it facilitates the training of deep networks through shortcut connections, which mitigates the problem of vanishing gradients. The ResNet18 is known for its balance between performance and computational efficiency.

**ViT: TinyViT**  Vision Transformer (ViT), introduced by Dosovitskiy et al. [19], adapts the transformer architecture to image classification by treating images as sequences of patches. We consider TinyViT from Wu et al. [48], as it is a compact version of ViT designed to be parameter-efficient while maintaining high performance.

# D  Privacy Evaluation

## D.1  Unlearning-Membership Inference Attack (U-MIA)

A common approach to evaluate the quality of unlearning methods is to attack the unlearned models with a form of Membership inference Attack (MIA). Membership Inference Attacks attempt to determine whether a specific data point was part of the model train data. The efficacy of the Membership Inference Attack has been used as a metric to evaluate the success of unlearning algorithms. A general approach to such an attack is as follows. Assume $f\theta$ is a trained model with parameters $\theta$, and let $\mathcal{L}$ be a loss function, such as the cross-entropy loss. Then, compute the losses for each sample from two sets of data $A$ and $B$ (of equal size) and train a binary classification model such as logistic regression with labels $y_i^A = 1$ for points $i$ in $A$ and $y_i^B = 0$ for points $i$ in $B$. An accuracy score from the classifier close to $1.0$ indicates that the classifier can perfectly distinguish between samples from $A$ and $B$ based on the loss values. A score of $0.5$ indicates that the ability to distinguish is close to random.

## D.2  Unlearning -Likelihood Ratio Attack (U-LiRA)

The performance of a general MIA can be improved by considering, *e.g.*, a per-sample attack such as LiRA [13, 31]. For any given point, we wish to determine whether the outputs from the unlearned models differ from those of models that have never seen the data point. To assess the attack robustly,

Table 4: MNIST - ResNet18

| unlearner | RA | FA | TA | RR | FR | TR | RetDev | Indisc | T-MIA | RTE |
|---|---|---|---|---|---|---|---|---|---|---|
| BT | 1.00 | 1.00 | 0.99 | 1.00 | 1.01 | 1.00 | 0.01 | 0.99 | 0.50 | 9.76 |
| CF-k | 1.00 | 1.00 | 0.99 | 1.00 | 1.01 | 1.00 | 0.01 | 0.98 | 0.51 | 5.86 |
| CFW | 1.00 | 1.00 | 0.99 | 1.00 | 1.00 | 1.00 | 0.00 | 1.00 | 0.50 | 4.29 |
| CT | 1.00 | 0.99 | 0.99 | 1.00 | 1.00 | 1.00 | 0.00 | 1.00 | 0.50 | 5.83 |
| EU-k | - | - | - | - | - | - | - | - | - | - |
| FCS | 1.00 | 0.99 | 0.99 | 1.00 | 1.00 | 1.00 | 0.00 | 1.00 | 0.50 | 2.11 |
| FF | - | - | - | - | - | - | - | - | - | 27.64 |
| FT | 1.00 | 0.99 | 0.99 | 1.00 | 1.00 | 1.00 | 0.00 | 0.99 | 0.51 | 5.07 |
| GA | 0.98 | 0.98 | 0.97 | 0.98 | 0.99 | 0.98 | 0.04 | 0.99 | 0.51 | 33.97 |
| IU | - | - | - | - | - | - | - | - | - | - |
| KDE | 1.00 | 0.99 | 0.99 | 1.00 | 1.00 | 1.00 | 0.00 | 1.00 | 0.50 | 3.27 |
| MSG | 1.00 | 0.99 | 0.99 | 1.00 | 1.00 | 1.00 | 0.00 | 1.00 | 0.50 | 4.29 |
| NG+ | 1.00 | 0.99 | 0.99 | 1.00 | 1.00 | 1.00 | 0.01 | 1.00 | 0.50 | 3.12 |
| O | 1.00 | 1.00 | 0.99 | 1.00 | 1.01 | 1.00 | 0.01 | 0.98 | 0.51 | 1.10 |
| PRMQ | 1.00 | 0.99 | 0.99 | 1.00 | 1.00 | 1.00 | 0.00 | 1.00 | 0.50 | 3.77 |
| R | 1.00 | 0.99 | 0.99 | 1.00 | 1.00 | 1.00 | 0.00 | 1.00 | 0.50 | 1.00 |
| RNI | 1.00 | 1.00 | 1.00 | 1.00 | 1.00 | 1.00 | 0.01 | 1.00 | 0.50 | 4.70 |
| SCRUB | - | - | - | - | - | - | - | - | - | - |
| SRL | 1.00 | 1.00 | 0.99 | 1.00 | 1.01 | 0.99 | 0.01 | 0.99 | 0.51 | 8.55 |
| SalUN | 1.00 | 1.00 | 0.99 | 1.00 | 1.00 | 0.99 | 0.01 | 0.99 | 0.50 | 3.19 |

we evaluate it across multiple models, using shadow models trained on various retain/forget sets. Specifically, we first train $n$ models based on $n$ splits of the training data. This train data is then split into 10 random retain and forget splits, and hence, we unlearn a total of $10n$ models. We then perform hyper-parameter sweeps, similar to what we do in the original results and unlearn using the optimal hyper-parameters, except that we consider $\frac{n}{2}$ sweeps and conduct 200 trials per sweep to determine the best hyper-parameters. In our setting, we set $n = 64$.

## E  Per dataset results

Here, we present the results for both ResNet18 and the TinyViT across datasets.

**ResNet18** We provide the tables with Retain Accuracy (RA), Forget Accuracy (FA), Test Accuracy (TA), Retain Retention (RR), Forget Retention (FR), Test Retention (TR), Performance Retention Deviation (RetDev), Indiscinerbility concerning the Test Set (Indisc), U-MIA on the Test set (T-MIA) and RunTime Efficiency (RTE) for every dataset using the ResNet18 model on MNIST (Table 4), FashionMNIST (Table 5), CIFAR-10 (Table 6), CIFAR-100 (Table 7) and UTKFace (Table 8). In general, CIFAR-100 provides the most visible differences, as the performance on the retain set is much higher than on the test. Datasets such as MNIST and FashionMNIST tend to show smaller differences between the methods as the performance on both the Retain and Test sets are similar, to begin with.

**TinyViT** We provide the tables with RA, FA, TA, RR, FR, TR, RetDev, Indisc, T-MIA and RTE for MNIST (Table 9), FashionMNIST (Table 10), CIFAR-10 (Table 11) and CIFAR-100 (Table 12) using the TinyViT model.

## F  Per architectures rankings

Here, we present the rankings across datasets for ResNet18 (Table 13) and TinyVit (Table 14). We note that some methods, such as RNI or NG+, are less efficient on the ViT architectures regarding Indiscernibility. However, methods such as SCRUB are less efficient regarding Retention Deviation on the ViT architecture.

Table 5: FashionMNIST - ResNet18

| Unlearner | RA | FA | TA | RR | FR | TR | RetDev | Indisc | T-MIA | RTE |
|---|---|---|---|---|---|---|---|---|---|---|
| BT | 1.00 | 0.96 | 0.92 | 1.00 | 1.03 | 0.99 | 0.04 | 0.98 | 0.51 | 13.57 |
| CF-k | 0.98 | 0.97 | 0.91 | 0.98 | 1.05 | 0.98 | 0.09 | 0.92 | 0.54 | 16.31 |
| CFW | 1.00 | 0.95 | 0.92 | 1.00 | 1.02 | 0.99 | 0.03 | 0.97 | 0.51 | 4.90 |
| CT | 1.00 | 0.92 | 0.92 | 1.00 | 0.99 | 0.99 | 0.02 | 0.99 | 0.50 | 4.47 |
| EU-k | - | - | - | - | - | - | - | - | - | - |
| FCS | 0.98 | 0.93 | 0.91 | 0.98 | 1.00 | 0.98 | 0.04 | 0.98 | 0.51 | 2.79 |
| FF | - | - | - | - | - | - | - | - | - | - |
| FT | 1.00 | 0.95 | 0.92 | 1.00 | 1.02 | 1.00 | 0.02 | 0.98 | 0.51 | 4.29 |
| GA | - | - | - | - | - | - | - | - | - | - |
| IU | 1.00 | 1.00 | 0.93 | 1.00 | 1.08 | 1.00 | 0.08 | 0.87 | 0.56 | 14.97 |
| KDE | 1.00 | 0.93 | 0.92 | 1.00 | 1.00 | 1.00 | 0.01 | 0.99 | 0.50 | 3.22 |
| MSG | 1.00 | 0.93 | 0.91 | 1.00 | 1.00 | 0.99 | 0.01 | 0.98 | 0.51 | 3.32 |
| NG+ | 0.99 | 0.94 | 0.91 | 0.99 | 1.01 | 0.99 | 0.03 | 0.99 | 0.49 | 3.21 |
| O | 1.00 | 1.00 | 0.93 | 1.00 | 1.08 | 1.00 | 0.08 | 0.87 | 0.56 | 1.11 |
| PRMQ | 0.98 | 0.93 | 0.91 | 0.98 | 1.00 | 0.99 | 0.04 | 0.98 | 0.51 | 3.77 |
| R | 1.00 | 0.93 | 0.92 | 1.00 | 1.00 | 1.00 | 0.00 | 1.00 | 0.50 | 1.00 |
| RNI | 0.98 | 0.93 | 0.91 | 0.98 | 1.00 | 0.98 | 0.04 | 0.98 | 0.51 | 3.25 |
| SCRUB | 0.95 | 0.93 | 0.90 | 0.95 | 1.00 | 0.98 | 0.08 | 0.97 | 0.51 | 5.62 |
| SRL | 1.00 | 0.97 | 0.92 | 1.00 | 1.04 | 1.00 | 0.05 | 0.98 | 0.51 | 24.85 |
| SalUN | 0.99 | 0.97 | 0.92 | 0.99 | 1.04 | 0.99 | 0.06 | 0.98 | 0.51 | 25.02 |

Table 6: CIFAR-10 - ResNet18

| Unlearner | RA | FA | TA | RR | FR | TR | RetDev | Indisc | T-MIA | RTE |
|---|---|---|---|---|---|---|---|---|---|---|
| BT | 0.94 | 0.87 | 0.84 | 0.94 | 1.00 | 0.96 | 0.10 | 0.97 | 0.48 | 51.96 |
| CF-k | - | - | - | - | - | - | - | - | - | - |
| CFW | 1.00 | 0.81 | 0.80 | 1.00 | 0.92 | 0.92 | 0.16 | 1.00 | 0.50 | 4.67 |
| CT | 1.00 | 0.82 | 0.81 | 1.00 | 0.93 | 0.93 | 0.14 | 0.99 | 0.50 | 17.49 |
| EU-k | - | - | - | - | - | - | - | - | - | - |
| FCS | 0.99 | 0.86 | 0.84 | 0.99 | 0.98 | 0.96 | 0.07 | 0.98 | 0.49 | 22.53 |
| FF | - | - | - | - | - | - | - | - | - | - |
| FT | 1.00 | 0.84 | 0.82 | 1.00 | 0.96 | 0.95 | 0.09 | 1.00 | 0.50 | 8.15 |
| GA | 0.91 | 0.89 | 0.81 | 0.91 | 1.02 | 0.93 | 0.18 | 0.92 | 0.54 | 91.48 |
| IU | 0.95 | 0.94 | 0.84 | 0.95 | 1.08 | 0.97 | 0.16 | 0.91 | 0.55 | 64.61 |
| KDE | 0.98 | 0.84 | 0.80 | 0.98 | 0.96 | 0.92 | 0.15 | 0.97 | 0.52 | 6.33 |
| MSG | 1.00 | 0.85 | 0.83 | 1.00 | 0.97 | 0.95 | 0.08 | 0.99 | 0.51 | 6.80 |
| NG+ | 0.97 | 0.89 | 0.85 | 0.98 | 1.02 | 0.97 | 0.07 | 0.98 | 0.51 | 12.89 |
| O | 0.96 | 0.96 | 0.85 | 0.96 | 1.10 | 0.98 | 0.16 | 0.89 | 0.55 | 1.08 |
| PRMQ | 1.00 | 0.86 | 0.83 | 1.00 | 0.98 | 0.95 | 0.07 | 0.98 | 0.51 | 4.93 |
| R | 1.00 | 0.87 | 0.87 | 1.00 | 1.00 | 1.00 | 0.00 | 1.00 | 0.50 | 1.00 |
| RNI | 1.00 | 0.83 | 0.81 | 1.00 | 0.95 | 0.93 | 0.12 | 0.99 | 0.50 | 3.60 |
| SCRUB | 0.99 | 0.85 | 0.85 | 0.99 | 0.97 | 0.98 | 0.07 | 0.99 | 0.50 | 2.57 |
| SRL | 0.99 | 0.93 | 0.84 | 0.99 | 1.06 | 0.97 | 0.10 | 0.98 | 0.49 | 5.52 |
| SalUN | 0.98 | 0.90 | 0.84 | 0.98 | 1.04 | 0.97 | 0.08 | 0.96 | 0.48 | 18.04 |

Table 7: CIFAR-100 - ResNet18. CIFAR-100 provides the most visible comparison as there is a large gap in performance between the Retain Set and Test set, this leads to much larger RetDev scores.

| Unlearner | RA | FA | TA | RR | FR | TR | RetDev | Indisc | T-MIA | RTE |
|---|---|---|---|---|---|---|---|---|---|---|
| BT | 0.98 | 0.68 | 0.54 | 0.98 | 1.25 | 0.99 | 0.27 | 0.95 | 0.48 | 9.39 |
| CF-k | 1.00 | 0.83 | 0.56 | 1.00 | 1.53 | 1.02 | 0.55 | 0.73 | 0.63 | 5.91 |
| CFW | 0.98 | 0.43 | 0.43 | 0.98 | 0.79 | 0.78 | 0.44 | 1.00 | 0.50 | 6.17 |
| CT | 0.99 | 0.53 | 0.53 | 0.99 | 0.97 | 0.97 | 0.07 | 0.99 | 0.49 | 11.82 |
| EU-k | - | - | - | - | - | - | - | - | - | - |
| FCS | 0.98 | 0.54 | 0.55 | 0.98 | 0.99 | 1.01 | 0.04 | 0.92 | 0.54 | 3.02 |
| FF | - | - | - | - | - | - | - | - | - | - |
| FT | 0.98 | 0.55 | 0.54 | 0.98 | 1.02 | 0.98 | 0.05 | 0.99 | 0.50 | 5.16 |
| GA | 0.34 | 0.33 | 0.24 | 0.34 | 0.60 | 0.44 | 1.61 | 0.90 | 0.55 | 39.97 |
| IU | - | - | - | - | - | - | - | - | - | - |
| KDE | 0.99 | 0.52 | 0.51 | 0.99 | 0.95 | 0.94 | 0.11 | 0.99 | 0.50 | 3.98 |
| MSG | 0.91 | 0.38 | 0.38 | 0.91 | 0.69 | 0.69 | 0.71 | 1.00 | 0.50 | 4.49 |
| NG+ | 0.89 | 0.59 | 0.49 | 0.89 | 1.08 | 0.89 | 0.29 | 0.98 | 0.49 | 12.14 |
| O | 0.98 | 0.98 | 0.56 | 0.98 | 1.81 | 1.02 | 0.85 | 0.53 | 0.73 | 1.10 |
| PRMQ | 0.97 | 0.47 | 0.46 | 0.97 | 0.86 | 0.85 | 0.32 | 1.00 | 0.50 | 4.34 |
| R | 1.00 | 0.55 | 0.55 | 1.00 | 1.00 | 1.00 | 0.00 | 0.99 | 0.49 | 1.00 |
| RNI | 0.99 | 0.45 | 0.45 | 0.99 | 0.83 | 0.82 | 0.36 | 0.98 | 0.49 | 3.65 |
| SCRUB | 0.97 | 0.50 | 0.53 | 0.97 | 0.91 | 0.96 | 0.15 | 0.98 | 0.51 | 3.81 |
| SRL | 1.00 | 0.55 | 0.52 | 1.00 | 1.00 | 0.95 | 0.06 | 0.98 | 0.49 | 3.67 |
| SalUN | 0.98 | 0.49 | 0.51 | 0.98 | 0.91 | 0.93 | 0.18 | 0.99 | 0.49 | 10.66 |

Table 8: UTKFace - ResNet18

| Unlearner | RA | FA | TA | RR | FR | TR | RetDev | Indisc | T-MIA | RTE |
|---|---|---|---|---|---|---|---|---|---|---|
| BT | 1.00 | 0.74 | 0.73 | 1.00 | 1.00 | 0.96 | 0.04 | 0.99 | 0.50 | 12.48 |
| CF-k | 1.00 | 1.00 | 0.75 | 1.00 | 1.34 | 0.99 | 0.35 | 0.70 | 0.65 | 5.35 |
| CFW | 1.00 | 0.76 | 0.76 | 1.00 | 1.02 | 1.00 | 0.02 | 1.00 | 0.50 | 5.54 |
| CT | 1.00 | 0.75 | 0.76 | 1.00 | 1.01 | 1.00 | 0.01 | 0.99 | 0.50 | 13.34 |
| EU-k | 0.72 | 0.61 | 0.59 | 0.72 | 0.82 | 0.77 | 0.68 | 0.99 | 0.51 | 11.42 |
| FCS | 0.90 | 0.70 | 0.70 | 0.91 | 0.94 | 0.93 | 0.23 | 0.99 | 0.50 | 4.33 |
| FF | - | - | - | - | - | - | - | - | - | - |
| FT | 1.00 | 0.76 | 0.77 | 1.00 | 1.02 | 1.01 | 0.04 | 1.00 | 0.50 | 5.78 |
| GA | 0.49 | 0.47 | 0.40 | 0.49 | 0.63 | 0.53 | 1.34 | 0.92 | 0.54 | 235.10 |
| IU | 1.00 | 1.00 | 0.76 | 1.00 | 1.34 | 1.01 | 0.35 | 0.62 | 0.69 | 33.77 |
| KDE | 0.99 | 0.79 | 0.76 | 0.99 | 1.06 | 1.00 | 0.07 | 0.97 | 0.52 | 8.19 |
| MSG | 1.00 | 0.80 | 0.76 | 1.00 | 1.08 | 1.00 | 0.08 | 0.96 | 0.52 | 7.57 |
| NG+ | 0.94 | 0.80 | 0.72 | 0.95 | 1.07 | 0.95 | 0.18 | 0.99 | 0.51 | 6.73 |
| O | 1.00 | 1.00 | 0.76 | 1.00 | 1.34 | 1.01 | 0.35 | 0.61 | 0.69 | 1.09 |
| PRMQ | 0.91 | 0.72 | 0.72 | 0.91 | 0.97 | 0.95 | 0.17 | 1.00 | 0.50 | 5.88 |
| R | 1.00 | 0.75 | 0.76 | 1.00 | 1.00 | 1.00 | 0.00 | 1.00 | 0.50 | 1.00 |
| RNI | 0.96 | 0.75 | 0.73 | 0.96 | 1.01 | 0.96 | 0.08 | 0.98 | 0.51 | 5.11 |
| SCRUB | 0.80 | 0.76 | 0.69 | 0.80 | 1.01 | 0.92 | 0.29 | 0.94 | 0.53 | 4.64 |
| SRL | 1.00 | 0.80 | 0.73 | 1.00 | 1.08 | 0.97 | 0.11 | 0.99 | 0.51 | 12.05 |
| SalUN | 0.97 | 0.79 | 0.73 | 0.98 | 1.06 | 0.96 | 0.12 | 0.97 | 0.52 | 36.80 |

Table 9: MNIST - TinyViT

| Unlearner | RA | FA | TA | RR | FR | TR | RetDev | Indisc | T-MIA | RTE |
|---|---|---|---|---|---|---|---|---|---|---|
| BT | 1.00 | 1.00 | 0.99 | 1.00 | 1.01 | 1.00 | 0.01 | 1.00 | 0.50 | 4.39 |
| CF-k | 1.00 | 1.00 | 0.99 | 1.00 | 1.01 | 1.00 | 0.01 | 0.99 | 0.51 | 123.88 |
| CFW | 1.00 | 0.99 | 0.99 | 1.00 | 1.00 | 1.00 | 0.00 | 0.99 | 0.50 | 5.15 |
| CT | 1.00 | 0.99 | 0.99 | 1.00 | 1.00 | 1.00 | 0.00 | 1.00 | 0.50 | 5.92 |
| EU-k | 1.00 | 1.00 | 0.99 | 1.00 | 1.01 | 1.00 | 0.01 | 0.99 | 0.50 | 11.93 |
| FCS | 1.00 | 1.00 | 0.99 | 1.00 | 1.01 | 1.00 | 0.01 | 0.99 | 0.49 | 7.42 |
| FF | - | - | - | - | - | - | - | - | - | - |
| FT | 1.00 | 0.99 | 0.99 | 1.00 | 1.00 | 1.00 | 0.01 | 1.00 | 0.50 | 7.69 |
| GA | 0.97 | 0.97 | 0.96 | 0.97 | 0.98 | 0.97 | 0.08 | 0.99 | 0.51 | 390.99 |
| IU | - | - | - | - | - | - | - | - | - | - |
| KDE | 1.00 | 0.99 | 0.99 | 1.00 | 1.00 | 1.00 | 0.01 | 1.00 | 0.50 | 5.35 |
| MSG | 1.00 | 0.99 | 0.99 | 1.00 | 1.00 | 1.00 | 0.00 | 1.00 | 0.50 | 5.21 |
| NG+ | 1.00 | 0.99 | 0.99 | 1.00 | 1.00 | 1.00 | 0.00 | 0.99 | 0.50 | 2.74 |
| O | 1.00 | 1.00 | 0.99 | 1.00 | 1.01 | 1.00 | 0.01 | 0.99 | 0.51 | 0.97 |
| PRMQ | 1.00 | 0.99 | 0.99 | 1.00 | 1.00 | 1.00 | 0.00 | 1.00 | 0.50 | 6.88 |
| R | 1.00 | 0.99 | 0.99 | 1.00 | 1.00 | 1.00 | 0.00 | 1.00 | 0.50 | 1.00 |
| RNI | 1.00 | 0.99 | 0.99 | 1.00 | 1.00 | 1.00 | 0.01 | 1.00 | 0.50 | 8.03 |
| SCRUB | 0.99 | 0.99 | 0.99 | 0.99 | 1.00 | 1.00 | 0.01 | 1.00 | 0.50 | 3.54 |
| SRL | 1.00 | 0.99 | 0.99 | 1.00 | 1.00 | 1.00 | 0.01 | 0.98 | 0.51 | 16.37 |
| SalUN | 1.00 | 1.00 | 0.99 | 1.00 | 1.01 | 0.99 | 0.01 | 0.99 | 0.50 | 43.67 |

Table 10: FashionMNIST - TinyViT

| Unlearner | RA | FA | TA | RR | FR | TR | RetDev | Indisc | T-MIA | RTE |
|---|---|---|---|---|---|---|---|---|---|---|
| BT | 0.97 | 0.94 | 0.91 | 0.97 | 1.01 | 0.99 | 0.04 | 0.98 | 0.51 | 3.48 |
| CF-k | - | - | - | - | - | - | - | - | - | - |
| CFW | 0.99 | 0.94 | 0.92 | 0.99 | 1.01 | 1.00 | 0.02 | 0.99 | 0.51 | 5.40 |
| CT | 0.98 | 0.92 | 0.91 | 0.98 | 0.99 | 0.99 | 0.04 | 0.99 | 0.50 | 6.00 |
| EU-k | 0.95 | 0.94 | 0.91 | 0.95 | 1.01 | 0.99 | 0.07 | 0.97 | 0.51 | 5.34 |
| FCS | 0.98 | 0.93 | 0.91 | 0.98 | 1.01 | 0.99 | 0.04 | 0.98 | 0.51 | 4.58 |
| FF | - | - | - | - | - | - | - | - | - | - |
| FT | 0.99 | 0.94 | 0.92 | 1.00 | 1.01 | 1.00 | 0.02 | 0.98 | 0.51 | 5.12 |
| GA | 0.92 | 0.91 | 0.85 | 0.92 | 0.99 | 0.93 | 0.17 | 0.93 | 0.53 | 50.72 |
| IU | - | - | - | - | - | - | - | - | - | - |
| KDE | 1.00 | 0.94 | 0.92 | 1.00 | 1.02 | 1.00 | 0.02 | 0.98 | 0.51 | 3.38 |
| MSG | 0.96 | 0.92 | 0.91 | 0.96 | 1.00 | 0.99 | 0.05 | 0.99 | 0.51 | 8.21 |
| NG+ | 0.97 | 0.92 | 0.91 | 0.97 | 1.00 | 0.99 | 0.05 | 0.98 | 0.51 | 13.65 |
| O | 1.00 | 1.00 | 0.92 | 1.00 | 1.08 | 1.00 | 0.08 | 0.89 | 0.56 | 0.97 |
| PRMQ | 0.98 | 0.94 | 0.91 | 0.98 | 1.01 | 0.99 | 0.04 | 0.98 | 0.51 | 4.67 |
| R | 1.00 | 0.93 | 0.92 | 1.00 | 1.00 | 1.00 | 0.00 | 1.00 | 0.50 | 1.00 |
| RNI | 0.97 | 0.94 | 0.91 | 0.97 | 1.01 | 0.99 | 0.05 | 0.97 | 0.51 | 5.00 |
| SCRUB | 0.96 | 0.95 | 0.91 | 0.96 | 1.03 | 0.99 | 0.08 | 0.96 | 0.52 | 9.56 |
| SRL | 0.98 | 0.94 | 0.91 | 0.98 | 1.01 | 0.99 | 0.04 | 1.00 | 0.50 | 9.41 |
| SalUN | 0.97 | 0.94 | 0.91 | 0.97 | 1.01 | 0.99 | 0.05 | 0.99 | 0.51 | 6.53 |

Table 11: CIFAR-10 - TinyViT

| Unlearner | RA | FA | TA | RR | FR | TR | RetDev | Indisc | T-MIA | RTE |
|---|---|---|---|---|---|---|---|---|---|---|
| BT | 0.91 | 0.91 | 0.85 | 0.91 | 1.02 | 0.97 | 0.14 | 0.99 | 0.50 | 4.13 |
| CF-k | 0.99 | 0.89 | 0.84 | 0.99 | 1.00 | 0.95 | 0.06 | 0.96 | 0.52 | 5.18 |
| CFW | 0.98 | 0.87 | 0.84 | 0.99 | 0.98 | 0.96 | 0.07 | 0.98 | 0.51 | 28.85 |
| CT | 0.98 | 0.82 | 0.81 | 0.98 | 0.93 | 0.92 | 0.17 | 1.00 | 0.50 | 23.48 |
| EU-k | 0.90 | 0.90 | 0.84 | 0.90 | 1.02 | 0.95 | 0.16 | 0.97 | 0.52 | 43.25 |
| FCS | 0.98 | 0.84 | 0.83 | 0.98 | 0.95 | 0.94 | 0.13 | 0.99 | 0.49 | 5.15 |
| FF | - | - | - | - | - | - | - | - | - | - |
| FT | 1.00 | 0.87 | 0.84 | 1.00 | 0.98 | 0.95 | 0.07 | 0.98 | 0.51 | 5.85 |
| GA | 0.85 | 0.85 | 0.80 | 0.85 | 0.96 | 0.91 | 0.29 | 0.97 | 0.52 | 514.77 |
| IU | - | - | - | - | - | - | - | - | - | - |
| KDE | 0.97 | 0.86 | 0.84 | 0.97 | 0.97 | 0.96 | 0.11 | 0.99 | 0.50 | 5.27 |
| MSG | 1.00 | 0.85 | 0.83 | 1.00 | 0.96 | 0.94 | 0.10 | 0.99 | 0.51 | 7.38 |
| NG+ | 0.93 | 0.86 | 0.85 | 0.93 | 0.97 | 0.96 | 0.14 | 0.99 | 0.50 | 4.10 |
| O | 0.92 | 0.92 | 0.86 | 0.92 | 1.04 | 0.97 | 0.15 | 0.95 | 0.53 | 0.97 |
| PRMQ | 1.00 | 0.87 | 0.84 | 1.00 | 0.98 | 0.95 | 0.07 | 0.99 | 0.51 | 4.00 |
| R | 1.00 | 0.89 | 0.88 | 1.00 | 1.00 | 1.00 | 0.00 | 1.00 | 0.50 | 1.00 |
| RNI | 0.97 | 0.84 | 0.81 | 0.98 | 0.95 | 0.92 | 0.15 | 0.98 | 0.51 | 6.50 |
| SCRUB | 1.00 | 0.84 | 0.84 | 1.00 | 0.95 | 0.95 | 0.10 | 0.99 | 0.50 | - |
| SRL | 0.97 | 0.88 | 0.84 | 0.97 | 0.99 | 0.96 | 0.08 | 0.99 | 0.49 | 8.52 |
| SalUN | 0.96 | 0.89 | 0.85 | 0.96 | 1.00 | 0.96 | 0.08 | 0.99 | 0.50 | 8.30 |

Table 12: CIFAR-100 - TinyViT

| Unlearner | RA | FA | TA | RR | FR | TR | RetDev | Indisc | T-MIA | RTE |
|---|---|---|---|---|---|---|---|---|---|---|
| BT | 0.82 | 0.66 | 0.57 | 0.82 | 1.11 | 0.96 | 0.33 | 0.94 | 0.53 | 31.63 |
| CF-k | 0.24 | 0.18 | 0.18 | 0.24 | 0.31 | 0.30 | 2.15 | 0.98 | 0.51 | 5.97 |
| CFW | 0.98 | 0.58 | 0.56 | 0.98 | 0.97 | 0.95 | 0.10 | 0.99 | 0.51 | 9.18 |
| CT | 0.97 | 0.55 | 0.55 | 0.98 | 0.93 | 0.93 | 0.17 | 0.99 | 0.49 | 9.80 |
| EU-k | 0.61 | 0.60 | 0.49 | 0.61 | 1.01 | 0.81 | 0.59 | 0.90 | 0.55 | 19.29 |
| FCS | 0.89 | 0.60 | 0.58 | 0.89 | 1.01 | 0.98 | 0.13 | 0.97 | 0.48 | 7.05 |
| FF | - | - | - | - | - | - | - | - | - | - |
| FT | 1.00 | 0.56 | 0.55 | 1.00 | 0.94 | 0.91 | 0.15 | 1.00 | 0.50 | 5.90 |
| GA | 0.60 | 0.58 | 0.46 | 0.60 | 0.97 | 0.77 | 0.66 | 0.87 | 0.56 | 91.91 |
| IU | - | - | - | - | - | - | - | - | - | - |
| KDE | 0.93 | 0.58 | 0.57 | 0.94 | 0.98 | 0.96 | 0.13 | 0.99 | 0.50 | 4.03 |
| MSG | 0.97 | 0.57 | 0.56 | 0.97 | 0.95 | 0.93 | 0.15 | 1.00 | 0.50 | 5.89 |
| NG+ | 0.84 | 0.57 | 0.55 | 0.84 | 0.95 | 0.92 | 0.29 | 0.96 | 0.52 | 3.01 |
| O | 0.87 | 0.87 | 0.61 | 0.87 | 1.46 | 1.02 | 0.61 | 0.74 | 0.63 | 0.98 |
| PRMQ | 0.95 | 0.62 | 0.57 | 0.95 | 1.03 | 0.96 | 0.13 | 0.95 | 0.52 | 5.58 |
| R | 1.00 | 0.60 | 0.60 | 1.00 | 1.00 | 1.00 | 0.00 | 1.00 | 0.50 | 1.00 |
| RNI | 0.85 | 0.52 | 0.51 | 0.85 | 0.87 | 0.85 | 0.42 | 0.99 | 0.50 | 5.38 |
| SCRUB | 0.77 | 0.64 | 0.57 | 0.77 | 1.08 | 0.96 | 0.35 | 0.93 | 0.53 | 6.17 |
| SRL | 0.98 | 0.57 | 0.57 | 0.98 | 0.96 | 0.96 | 0.11 | 0.97 | 0.49 | 5.92 |
| SalUN | 0.97 | 0.58 | 0.57 | 0.97 | 0.97 | 0.96 | 0.10 | 0.98 | 0.49 | 7.03 |

Table 13: Ranking on ResNet

| | | Retention Deviation | | | | | | Indiscernibility | | | |
|---|---|---|---|---|---|---|---|---|---|---|---|
| Rank | Method | G1 | G2 | G3 | F | Rank | Method | G1 | G2 | G3 | F |
| 1 | FT | 5 | 0 | 0 | 0 | 1 | CFW | 5 | 0 | 0 | 0 |
| 2 | FCS | 4 | 1 | 0 | 0 | 1 | CT | 5 | 0 | 0 | 0 |
| 2 | MSG | 4 | 1 | 0 | 0 | 1 | MSG | 5 | 0 | 0 | 0 |
| 3 | CT | 4 | 0 | 1 | 0 | 1 | RNI | 5 | 0 | 0 | 0 |
| 3 | KDE | 4 | 0 | 1 | 0 | 2 | FT | 4 | 1 | 0 | 0 |
| 4 | NG+ | 3 | 2 | 0 | 0 | 2 | KDE | 4 | 1 | 0 | 0 |
| 4 | PRMQ | 3 | 2 | 0 | 0 | 2 | NG+ | 4 | 1 | 0 | 0 |
| 4 | SalUN | 3 | 2 | 0 | 0 | 2 | PRMQ | 4 | 1 | 0 | 0 |
| 5 | CFW | 3 | 1 | 1 | 0 | 3 | FCS | 3 | 2 | 0 | 0 |
| 6 | SCRUB | 3 | 0 | 1 | 1 | 3 | SRL | 3 | 2 | 0 | 0 |
| 7 | SRL | 2 | 3 | 0 | 0 | 3 | SalUN | 3 | 2 | 0 | 0 |
| 8 | BT | 1 | 4 | 0 | 0 | 4 | SCRUB | 3 | 1 | 0 | 1 |
| 8 | RNI | 1 | 4 | 0 | 0 | 5 | BT | 2 | 3 | 0 | 0 |
| 9 | CF-k | 1 | 2 | 1 | 1 | 6 | EU-k | 1 | 0 | 0 | 4 |
| 10 | IU | 1 | 0 | 2 | 2 | 7 | GA | 0 | 3 | 1 | 1 |
| 11 | EU-k | 0 | 1 | 0 | 4 | 8 | CF-k | 0 | 1 | 3 | 1 |
| 12 | GA | 0 | 0 | 4 | 1 | 9 | IU | 0 | 0 | 3 | 2 |
| 13 | FF | 0 | 0 | 0 | 5 | 10 | FF | 0 | 0 | 0 | 5 |

Table 14: Ranking of ViT

| | | Retention Deviation | | | | | | Indiscernibility | | | |
|---|---|---|---|---|---|---|---|---|---|---|---|
| Rank | Method | G1 | G2 | G3 | F | Rank | Method | G1 | G2 | G3 | F |
| 1 | CFW | 4 | 0 | 0 | 0 | 1 | CT | 4 | 0 | 0 | 0 |
| 1 | MSG | 4 | 0 | 0 | 0 | 1 | MSG | 4 | 0 | 0 | 0 |
| 1 | PRMQ | 4 | 0 | 0 | 0 | 2 | KDE | 3 | 1 | 0 | 0 |
| 2 | CT | 3 | 1 | 0 | 0 | 2 | SalUN | 3 | 1 | 0 | 0 |
| 2 | FT | 3 | 1 | 0 | 0 | 3 | SRL | 3 | 0 | 1 | 0 |
| 2 | KDE | 3 | 1 | 0 | 0 | 4 | BT | 2 | 2 | 0 | 0 |
| 2 | SRL | 3 | 1 | 0 | 0 | 4 | CFW | 2 | 2 | 0 | 0 |
| 2 | SalUN | 3 | 1 | 0 | 0 | 4 | FCS | 2 | 2 | 0 | 0 |
| 3 | FCS | 2 | 2 | 0 | 0 | 4 | FT | 2 | 2 | 0 | 0 |
| 3 | NG+ | 2 | 2 | 0 | 0 | 4 | PRMQ | 2 | 2 | 0 | 0 |
| 4 | BT | 1 | 3 | 0 | 0 | 4 | RNI | 2 | 2 | 0 | 0 |
| 4 | RNI | 1 | 3 | 0 | 0 | 4 | SCRUB | 2 | 2 | 0 | 0 |
| 4 | SCRUB | 1 | 3 | 0 | 0 | 5 | NG+ | 1 | 3 | 0 | 0 |
| 5 | CF-k | 1 | 1 | 1 | 1 | 6 | CF-k | 1 | 1 | 1 | 1 |
| 6 | EU-k | 0 | 4 | 0 | 0 | 7 | EU-k | 0 | 2 | 2 | 0 |
| 7 | GA | 0 | 1 | 3 | 0 | 8 | GA | 0 | 1 | 3 | 0 |
| 8 | FF | 0 | 0 | 0 | 4 | 9 | FF | 0 | 0 | 0 | 4 |
| 8 | IU | 0 | 0 | 0 | 4 | 9 | IU | 0 | 0 | 0 | 4 |

# G  L2 Distances between model weights.

The distance between the Unlearned and Retrained models has also been considered in the literature to evaluate MU. Nevertheless, we observe that models end up at a similar distance to the Retrained model, with significant differences in performance. We further note that one challenging aspect of the L2 distance comparison is the different factors of Weight Decay used by the MU method. The hyper-parameter searches determine these Weight Decay factors, which can significantly vary from one unlearning method to another, making it challenging to compare methods. Furthermore, the best-performing method, MSG, is usually at the same distance as both the Original and Retrained

713 model. For each method, for each initialization seed, we computed the L2 distance between the
714 unlearned model $f_U$ and the retrained model $f_R$, as well as between the $f_U$ and $f_O$ (Figure 2).

715 Although having the same weight as the Retrained model would indicate that the unlearned model has
716 unlearned $\mathcal{D}_F$, our evaluations show that distance to the Retrained model might not be an adequate
717 evaluation metric for MU.

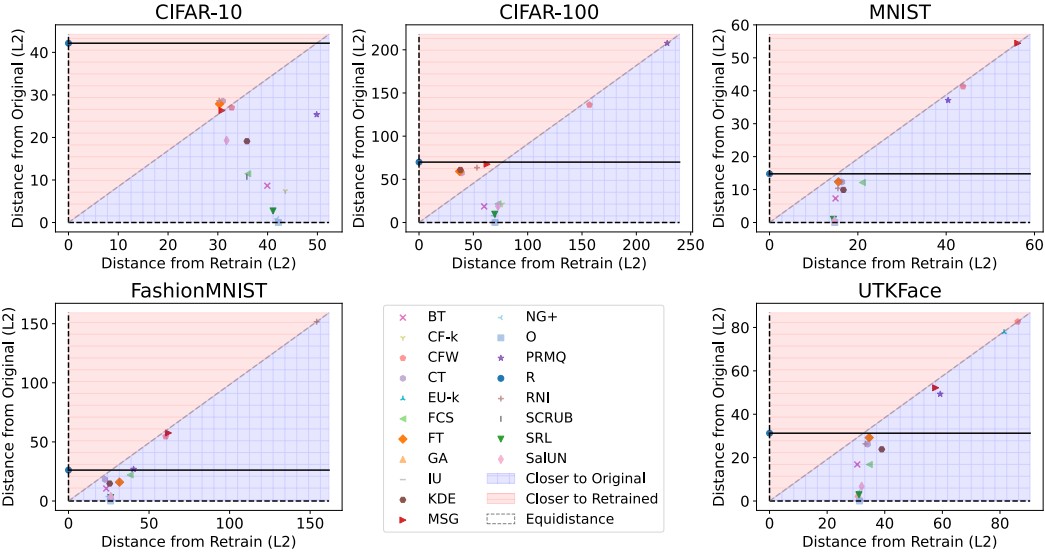

Figure 2: L2 Distance between the Unlearned ResNet18 models, the Original and Retrained models.
None of the unlearned models gets close to the Retrained model's weights; most unlearned Models
are closer to the Original model than the Retrained model.

# H  Requirements

719 We ran the experiments on compute clusters with different capacities. Nonetheless, each method was
720 tested on devices with the same specifications when recording run times: 1 NVIDIA L4 24GB GPU
721 and 4 Intel(R) Xeon(R) CPU @ 2.20GHz.