# OpenReview forum: "Deep Unlearn: Benchmarking Machine Unlearning"
_NeurIPS.cc/2024/Datasets_and_Benchmarks_Track — Submitted to NeurIPS 2024 Track Datasets and Benchmarks_

### Official Review · Reviewer_sLqq · 2024-07-09
**Important topic but with insufficient results and incomplete work.**

**Rating:** 3
**Confidence:** 5

**Review:**

1. The authors did not provide any justification for the paper’s checklist.
2. In [r1](Fisher forgetting), it worth noting that this method works under
strong assumption, such as stability of training algorithm (see its
Section 2.1 and Section 6), which is commonly ignored by most MU
papers that use it as the baseline method. Due to this, [r1] pretrains the
model in their experiments. So a concern is that directly comparing it
with other methods may lead to unfair comparison in the paper.
3. There are many other unlearning methods proposed in published
papers. however, the paper only includes 6 of them.
4. This paper does not evaluate robustness across different unlearning
budgets, which is a critical aspect in benchmarking work. The
experiments are limited to a 10% unlearning budget. Insights into the
robustness of various unlearning methods across different unlearning
budgets would enhance the study.
5. In [r2], it is noted that using random forgetting data for evaluating unlearning methods results in significant variance. However, this paper
only focuses on random forgetting data, which may lead to biased
evaluations and benchmarks.
6. In Section 4, the ranking evaluation details are unclear. How are the
conditions for G1, G2 or G3 defined? For a Failed group, what
specifically does “usable” refer to?
7. Why are there missing values from Table 4 to Table 12 for some
methods? Please explains these gaps in the texts.

[r1] Golatkar A, Achille A, Soatto S. Eternal sunshine of the spotless net: Selective
forgetting in deep networks[C]//Proceedings of the IEEE/CVF Conference on
Computer Vision and Pattern Recognition. 2020: 9304-9312.

[r2] Fan C, Liu J, Hero A, et al. Challenging forgets: Unveiling the worst-case forget
sets in machine unlearning[J]. arXiv preprint arXiv:2403.07362, 2024.

**Strengths:**

1. This paper evaluates 18 different Machine Unlearning methods on a variety datasets and models.
2. It investigates the reliability of commonly used MU baselines and MU
methods across datasets, models, and initializations.
3. Extensive experiments are conducted to validate the claims made in the paper.

**Additional Feedback:**

NA

**Clarity:**

The overall writing is clear although there are some minor grammatical errors
and typos.

**Correctness:**

Most claims made in the paper seem sound. However, there are still some
confusion needs to be made clear.
1. On page 2 line 49, the “8 recently published paper” is kind of confusing.
CF-k and EU-k come from the same paper, as do SCRUB and NG+. So I’m
thinking if it’s supposed to be 8 methods from 6 published paper.
2. On page 2, line 51 and page 4, line 161, the paper discusses influence
unlearning that it includes in both of these two places. However, it cites [33,
35] in the former and [33,47,34] in the latter and mention it adopts methods
from [34]. Addressing this inconsistency would improve the paper.
3. What does “O” in Figure 1 denote? There is no mention of this symbol
either in the caption or paper. Also in the Appendix Table 4, there is no
explanation of “R”.

**Documentation:**

This paper does not include new dataset.

**Ethics:**

No, there are no or only very minor ethics concerns.

**Limitations:**

The limitations are not discussed. The authors may want to cover the potential societal impact of the work in the paper.

**Opportunities For Improvement:**

See Review section.

**Relation To Prior Work:**

The paper does not either provide relations to prior work or mention there is
currently no related work of benchmarking machine unlearning.

**Summary And Contributions:**

Machine unlearning aims to remove the influence of specific data points from
a trained machine learning model, which is crucial for data privacy,
trustworthiness, and safety. This paper benchmarks the effectiveness of state-
of-the-art machine unlearning methods by evaluating 18 techniques across 5
benchmark datasets and 2 different model architectures, providing a
comprehensive assessment.

---

> ### Author Rebuttal · Authors · 2024-08-17
>
> Thank you for your feedback and suggestions.
>
> > 1. There are many other unlearning methods proposed in published papers. however, the paper only includes 6 of them.
>
> We selected these 6 unlearning methods because recent literature has identified them as some of the most promising approaches.
> In addition to these published methods, we also included and tested methods that, although not associated with published papers, 1) provide readily usable open-source code, which allows for reproducibility and broader community engagement, and 2) Their strong empirical performance in the competition, demonstrating their effectiveness in practice.
> We believe this selection strikes a balance between covering established methods and including high-performing approaches that may not yet have extensive publication records but are nonetheless valuable to consider.
>
> > 2. This paper does not evaluate robustness across different unlearning budgets, which is a critical aspect in benchmarking work. The experiments are limited to a 10% unlearning budget. Insights into the robustness of various unlearning methods across different unlearning budgets would enhance the study.
>
> We appreciate that evaluating robustness across different unlearning budgets is a valuable aspect of unlearning research. While we did not investigate robustness across various unlearning budgets in this study, we focused on evaluating multiple unlearning methods across diverse datasets, architectures, and hyperparameters. This approach allowed for a thorough analysis of these methods under a fixed unlearning budget, which the community has already considered.
> It's important to note that a fair evaluation of the effects of different unlearning budget sizes would require repeating the experiments across all our 100K models for each new budget size. Due to its significant time and computational resource requirements, this process was not feasible for this study. Conducting a limited evaluation to show only a small ablation study would not, in our view, add substantial value to the paper.
> We recognize the importance of exploring this aspect and plan to investigate robustness more deeply across different unlearning budgets for future work.
>
> > 3. No evaluation of tasks other than image classification or domains outside of vision. For example, the authors could consider natural language settings that involve private or sensitive data as a setting where machine unlearning would be critical.
>
> We appreciate the suggestion to explore tasks and domains beyond image classification, particularly in fields like natural language processing, where machine unlearning is vital for managing private or sensitive data.
> In the final version of the paper, we plan to provide a more detailed discussion of the limitations of our study, specifically addressing the challenges of generalizing our results to different models and data types. While we recognize the importance of extending our work to other domains, it is difficult to make broad extrapolations due to the unique inductive biases of different architectures and the varying properties of different data modalities.
> Additionally, we had to balance the depth of our analysis on each method with the breadth of the domains and settings we could explore. Conducting extensive hyperparameter selection across multiple datasets is both computationally and time-intensive, which limited our ability to extend the study to other domains in this initial work.

---

> > ### Comment · Reviewer_sLqq · 2024-08-23
> >
> > Thanks for the response. I feel my concerns are not handled responsibly.  For example, Response 3 is never mentioned by me. The methods are rather limited, looking like an extended version of [a].
> >
> > [a] Kurmanji, Meghdad, Peter Triantafillou, Jamie Hayes, and Eleni Triantafillou. "Towards Unbounded Machine Unlearning." NeurIPS 2023.

---

> > > ### Author Rebuttal · Authors · 2024-08-26
> > >
> > > I wanted to share some additional thoughts that might help clarify a few points:
> > > - Regarding point 2. on the comparison with other methods, our goal was to ensure a balanced evaluation. We optimized each method's hyperparameters with the same resources to achieve a fair and accurate performance comparison.
> > > - Concerning the use of random forgetting sets, our intention was not to approach from an adversarial standpoint but to provide a comprehensive evaluation of unlearning methods across a broad spectrum of hyperparameters, model configurations, and datasets.
> > > - The criteria for selecting G1, G2, and G3 are detailed in Section 4, where we describe the cut-offs from agglomerative clustering.
> > > - For missing values, particularly concerning the F group, this is clarified in the caption: "If a method does not produce any usable models, it is assigned to a failed group".
> > > - Regarding the comment on the methods being limited, while this is challenging to fully address without more specifics, we would like to highlight that we included several novel methods (beyond those in [a]), conducted an extensive hyperparameter search for each method, and compared results across various model configurations. This effort involved training approximately 100,000 models, which, to the best of our knowledge, constitutes the most comprehensive comparison of MU methods to date.
> > > I hope this provides more clarity and addresses your concerns. Please let us know if there are any other areas where we could provide further information.

---

### Official Review · Reviewer_K9a2 · 2024-07-15

**Rating:** 4
**Confidence:** 4
**Correctness:** Yes.
**Clarity:** Yes.

**Review:**

Evaluating machine unlearning approaches in a principled manner is an important open-problem. While the authors study a large number of approaches, the experimental setting is quite limited in both realism, models considered, and breadth of tasks. While the authors do offer some interesting insights based on the setting studied, I find the overall impact of the work is quite limited and find the connection to prior work also limited. I encourage authors to consider whether this work can be extended to more realistic settings and whether it can be repurposed to potentially serve as a standard benchmark for machine unlearning to increase its utility to the research community.

**Strengths:**

- This work studies the increasingly important topic of machine unlearning, a field of growing importance given recent regulations and broad applications of AI systems trained on sensitive data.
- The work provides a principled empirical study of existing methods for machine unlearning
- The authors conduct carefully designed experiments that account for computational efficiency, robustness to several evaluation approaches, and considers performance across seeds.

**Additional Feedback:**

N/A

**Documentation:**

N/A

**Ethics:**

No.

**Limitations:**

Yes, the authors describe several key limitations.

**Opportunities For Improvement:**

# Realism of evaluation setting
While the motivating problem is important, I find the experimental setting quite limited in how well it captures how machine unlearning methods would be useful in real world use cases. Specifically,
- Limited set of toysih image classification benchmarks with low resolution (except for UTK face) and limited number of very simple classes.
- No evaluation of pretrained foundation models that are becoming ubiquitous in practice. ResNet18 and TinyViT are very small architectures that are not representative of typical models used in practice where machine unlearining would be important.
- No evaluation of tasks other than image classification or domains outside of vision. For example, the authors could consider natural language settings that involve private or sensitive data as a setting where machine unlearning would be critical.

I find the experimental overall to be quite limited in supporting the claims made in this work as how machine unlearning approaches fair overall. To support such conclusions, much more comprehensive and realistic experimental settings should be considered.

# Limited Set of Takeaways and Impact Beyond The Current Findings
- I find the results, while insightful, have a somewhat limited impact as they mostly rehash what the public results of NeurIPS challenge. I agree the authors do a fantastic job of carefully comparing methods across dimensions, but find overall the takeaways in this work not drastically novel in comparison  to the already public findings of the NeurIPS Challenge. Perhaps the authors can highlight some of the differences more clearly?
- What now? Beyond the existing findings, it's not clear to me how useful this work would to the community more broadly. For example, the field has yet to converge on a reliable benchmark for machine unlearning. Are there learnings here that could be used to establish a benchmark for comparing new machine unlearning methods?

**Relation To Prior Work:**

There much more opportunity for the authors to connect the findings to prior work including results of the NeurIPS competition, and the memorization literature more broadly.

**Summary And Contributions:**

This work evaluates approaches for machine unlearning, the goal of which is to forget specific data points seen during training. The work benchmarks 18 machine unlearning methods including standard approaches such as finetuning and gradient ascent as well as top ranking methods from a recent NeurIPS machine unlearning challenge. The authors evaluate both ResNet18 and TinyViT architectures across 5 image classification datasets: MNIST, FashionMNIST, CIFAR10/100, and UTK-Face. The authors fix a forget set size of 10%. The authors show finetuning with proper hyperparameter tuning is a strong baseline and that Masked-Small Gradients is a promising approaches among the set of methods evaluated.

---

> ### Author Rebuttal · Authors · 2024-08-17
>
> Thank you for your feedback and suggestions regarding the scope of our evaluation, we address the following 3 comments:
>  > 1) Limited set of toyish image classification benchmarks with low resolution (except for UTK face) and limited number of very simple classes.
>
> > 2) No evaluation of pretrained foundation models that are becoming ubiquitous in practice. ResNet18 and TinyViT are very small architectures that are not representative of typical models used in practice where machine unlearning would be important.
>
> > 3) No evaluation of tasks other than image classification or domains outside of vision. For example, the authors could consider natural language settings that involve private or sensitive data as a setting where machine unlearning would be critical.
>
> Our primary focus in this study was conducting a comprehensive analysis of unlearning methods. We performed extensive hyperparameter tuning and evaluations across many models and datasets to ensure a robust understanding of each method's performance. Given this work's computational and time-intensive nature, we made deliberate trade-offs to prioritize the depth of our analysis over the breadth of settings.
>
> Regarding the choice of datasets, we selected datasets that are widely accessible to the broader research community. While these datasets may have simpler classes and lower resolutions, they are standard benchmarks that allow for reproducibility and comparability with other studies in the field. We believe that using these accessible datasets ensures that others can extend our results in the community.
> Concerning the use of smaller architectures like ResNet18 and TinyViT, we agree that evaluating unlearning methods on larger, pretrained foundation models is important for understanding their practical applicability. However, our intent was to focus on the unlearning mechanisms themselves. We assumed that methods unable to perform well on smaller, controlled models would likely struggle with more complex architectures as well. Moreover, the resource constraints associated with evaluating large-scale models across extensive hyper-parameter spaces further influenced our decision. Finally, we stress that the evaluated architectures are very widely used in practical image classification use-cases. We believe this provides a solid foundation for understanding the core principles of machine unlearning, which can be extended to other domains in future work.

---

> > ### Comment · Reviewer_K9a2 · 2024-08-19
> >
> > Thank you for the response. I appreciate some of the differences you highlight in the main response relative to the NeurIPS competition. Beyond the lack of reliability of the best performing models across architectures/hyperparameters, can you comment on any other scientific distinct findings from the work we may have missed relative to the competition?

---

> > > ### Author Response · Authors · 2024-08-21
> > >
> > > We really appreciate your prompt feedback.
> > >
> > > Please note that when we submitted our paper, the competition organizers had not yet released their analysis of the competition's findings and results and the only information available to us and the community was the ranking of various methods on the Kaggle website. While the competition organizers have published on arXiv on June 13, 2024 a paper on the findings from the competition (https://arxiv.org/abs/2406.09073), this was after our submission deadline.
> > >
> > > A key goal of our paper was to provide an independent analysis of the methods proposed in the competition, assess these in a more general setup across hyperparameter configurations, model weights and datasets, and in addition compare them to recent state-of-the-art approaches. The “scientific distinct findings” in our work are still the five key differences mentioned in our rebuttal: “Distinction from the NeurIPS Unlearning Competition” which we believe are very valuable to the community.

---

### Official Review · Reviewer_CWjQ · 2024-07-16
**The paper makes a significant contribution to the field of machine unlearning by providing a comprehensive and detailed benchmarking study.**

**Rating:** 6
**Confidence:** 3
**Clarity:** The paper is clearly written and well…

**Review:**

**Quality.** The paper is of high quality, presenting a comprehensive and detailed benchmarking of MU methods. The methodology includes a large-scale evaluation across multiple datasets and models.

**Clarity.** The paper is clearly written and well-organized.

**Originality.** The work is original in its extensive benchmarking of MU methods.

**Significance.** The findings are significant, particularly in identifying the top-performing MU methods and emphasizing the need for better baselines and hyperparameter selection.

**Pros.**
- Comprehensive evaluation of 18 SOTA MU methods
- Large-scale evaluation involving over 100,000 models
- Identification of top-performing methods (MSG and CT)
- Hyperparameter selection
- Clear presentation

**Cons.**
- The paper could include more discussion on the potential limitations of the study (see questions and suggestions below)
- While the evaluation is extensive, the general validity of the findings (for other architectures / datasets) could be further explored

**Strengths:**

- The study addresses an important problem related to data privacy and the right to be forgotten.

- The study provides a comprehensive evaluation that highlights the strengths and weaknesses of various approaches.

- The findings are highly relevant to the broader research community, particularly those working on data privacy, model trustworthiness, and safety.

- The research is well-executed and includes a comprehensive evaluation.

**Additional Feedback:**

See questions and suggestions above.

**Correctness:**

The claims made in the submission look correct. The paper provides a comprehensive evaluation of various MU methods using meaningful experimental design and appropriate evaluation metrics. The experiments are performed with detailed attention to hyperparameter tuning and baseline comparisons. The conclusions drawn from the results are supported by the data presented.

**Documentation:**

For benchmarks, the paper includes sufficient details. The datasets used in the study are well-documented, and the experimental setup is described in detail, including the neural network architectures, data augmentations, hyperparameters, and evaluation metrics. The authors promise to share the code of their experiments.

**Ethics:**

There are no ethical concerns with the submission.

**Limitations:**

The authors list limitations of their work. The study includes a thorough evaluation of various MU methods and discusses their performance and applicability. However, it would benefit from a more detailed exploration of potential biases in the datasets, provide more details on the ranking of the methods, and how the results could be extended to other types of models and data.

**Opportunities For Improvement:**

- Consider including a comparison table for the SOTA MU methods.

- The paper could benefit from a more in-depth discussion of its limitations, including how the results generalize to other types of models and data.

- Consider including additional benchmarks, especially those used in other domains. This could provide a more comprehensive understanding of the relative performance of the MU methods.

- line 128 - for how many epochs? How do the MU methods compare with respect to the speed of recovering the unlearned datapoint?

- line 189 - how was the forget set sampled? Uniformly? How would a non-uniform sampling affect the relative performance of the methods?

- Ranking method performance using clustering is generally highly appreciated, yet hides the magnitude of the differences in performance. The results in the appendix show that for some datasets these difference are very small. This raises a question whether the tables in the main paper could also be extended to show the distance between groups or similar.

**Relation To Prior Work:**

The paper provides a thorough review of related work, clearly discussing how this study differs from and builds upon previous contributions.

**Summary And Contributions:**

The paper presents a rigorous study on MU for DNNs. The authors evaluate 18 state-of-the-art MU methods across various benchmark datasets and models, with each evaluation conducted over 10 different initializations, amounting to an extensive evaluation involving over 100,000 models. The paper highlights the importance of proper baselines and hyperparameter selection, showing that methods like Masked Small Gradients and Convolution Transpose consistently outperform others in terms of model accuracy and run-time efficiency. The study is assessed using population-based membership inference attacks and per-sample unlearning likelihood ratio attacks.

---

> ### Author Rebuttal · Authors · 2024-08-17
>
> Thank you for your detailed review and valuable feedback.
>
> > 1. The paper could benefit from a more in-depth discussion of its limitations, including how the results generalize to other types of models and data.
>
> We appreciate your suggestion regarding the discussion of limitations. In the final version of the paper, we will include a more detailed discussion of the limitations. Specifically, we will expand on the challenges in generalizing our results to other models and data types.
> We acknowledge that generalizing our results to other data types or models would indeed be beneficial. However, it is challenging to make broad extrapolations because different architectures may rely on distinct inductive biases, and different modalities or types of data often exhibit varying properties. For example, in audio data and sequential data more generally, the interdependence and context across data points create additional complexities that make unlearning more difficult.
> That said, we have carefully chosen the architectures and data types used in our experiments to be representative of a significant portion of the domain. This approach provides a strong foundation for our findings, even as we remain aware of the specific limitations discussed above.

---

### Official Review · Reviewer_T7vM · 2024-07-24
**Deep Unlearn Benchmark: a solid effort to make benchmarking of unlearning methods more comprehensive**

**Rating:** 8
**Confidence:** 4
**Clarity:** The paper is well written and clear.

**Review:**

The paper is very well structured and well written. It is clear, has a strong motivation, and presents a comprehensive analysis that is bound to provide significant insights to the community for future work.

**Strengths:**

* The paper has a clear motivation, easy to follow descriptions, and is well written and structured.
* The choice of evaluation measures is meaningful for the given task of approximate unlearning and quite comprehensive. Respectively, many state of the art methods have been included in addition to intuitive baselines for the analysis to be valuable.
* The evaluation it self is interesting and it is nice to see an investigation that goes beyond “best in bold” in complicated tables with arbitrary absolute values. The grouping and ranking presented in the tables helps contextualize methods across a variety of datasets and thus extract valuable insights for the readers.
* It is expected that the drawn conclusions will fuel further investigation on larger datasets and helps steer directions of development of future methods.

**Additional Feedback:**

It would be great if the authors could upload their documented code with licensing and maintenance ideas in some form in the rebuttal phase for reviewers to look at (can be anonymous)

**Correctness:**

I could not find any incorrect statements. The analysis follows various prior works

**Documentation:**

Documentation in the appendix pdf is sound, but the hosting, licensing, maintenance, code aspects are hard to gauge without it being provided for review.

**Ethics:**

No ethical issues.

**Limitations:**

Limitations are appropriately discussed and valuable. Future work and impact are also added in a sufficient manner.

**Opportunities For Improvement:**

* Given that the page limit is 9 pages and almost the entirety of page 9 is left unused, it would be great to pull some content from the appendix into the main body. The first thing that comes to mind is adding at least one of the detailed result tables in the main body, and referring to the rest being in the appendix. As it stands right now, the main body only contains the relative ranking results, which is sufficient for grouping and analyzing results, but lacks absolute interpretation.
* It would have been nice to have some form of documentation of the source code, or the source code being provided in the appendix instead. Given that the track explicitly allows for this information to be uploaded (non-anonymously) it is hard to gauge whether documentation is sufficient. Given the rest of the paper and the supplemental pdf, I expect that it will be public and sufficiently evaluated and give the benefit of doubt.

**Relation To Prior Work:**

The relation to prior work is well done wrt methods and general set-up. The description could still be improved given that almost a full page of space is remaining. Specifically, it would be nice to have some more mention of other unlearning benchmarks and why they are lacking in certain ways (e.g. UnlearnCanvas and similar) .

**Summary And Contributions:**

The paper benchmarks  deep unlearning methods in a comprehensive and fairly exhaustive fashion. The inspiration is drawn from approximate unlearning and as such evaluation is based on e.g. membership inference attacks, among others, for a variety of evaluated methods and datasets.

---

> ### Author Rebuttal · Authors · 2024-08-17
>
> We are very thankful for your constructive feedback and comments.
>
> As mentioned in the general comment we are preparing the code for its public release.

---

### Author Rebuttal · Authors · 2024-08-17

We thank the reviewers for their constructive feedback and suggestions.

**Code**: The source code is under preparation to make sure that the community can easily develop on top of it. We will publish it by the conference date.

**Distinction from the NeurIPS Unlearning Competition**: The NeurIPS Competition has been a significant step forward. However, we have addressed some of the limitations, as follows:
A) The challenge relied on CASIA-SURF, a dataset not readily available to the public. Our benchmark relies on four datasets that are readily available to the public through PyTorch and one that is openly accessible.
B) The challenge relies on a single, fixed configuration of the model's weights, meaning the performance of the unlearning method is tied to this specific setup. This raises the question of whether the best-performing methods might be overfitting to this configuration. To ensure a more robust evaluation, we train multiple models with different initialization seeds and then apply the unlearning process to these varied weight configurations.
C) Unless implemented by participants, the competition did not compare against methods published in contemporary research papers.
D) Our evaluation settings show that the best-performing method from the competition is less reliable across different architectures and datasets.
E) Since unlearning methods can be sensitive to the choice of hyper-parameters, our benchmark ensures that all the methods are compared as fairly as possible using a search of 300 configurations of hyper-parameters (3 searches of 100 combinations), with each search starting from a model with different model weights. In the competition, this can differ from one submission to another.

**Why not more methods and datasets?** Due to computing limitations, we faced the choice of evaluating more unlearned methods and model architectures but using only a limited number of evaluations or extensively evaluating a select set of architectures and top-performing MU methods. Here we decided to focus on the latter for two reasons: 1) an extensive robustness analysis of MU methods across model configurations and MU hyperparameters is missing, and 2) Evaluating more methods would have resulted in limited ablation studies, from which drawing robust conclusions would have been challenging,

Our choice of architectures and datasets is based on what the community uses in other MU papers, and what is used in practice (we note here that while LLMs have taken the world by storm, millions of devices and use cases still rely on CNNs). To name a few comparison papers: “Model Sparsity Can Simplify Machine Unlearning” (Jia et al., 2023) uses CIFAR10, SVHN, CIFAR100 and ImageNet and studies ResNet18; “Challenging Forgets: Unveiling the Worst-Case Forget Sets in Machine Unlearning” (Fan et al., 2024) uses CIFAR10, ResNet18, CIFAR100, CelebA and Tiny Imagenet; “Towards reliable empirical machine unlearning evaluation: a game-theoretic view” (Tu et al., 2024) also uses CIFAR10 and ResNets.
For similar reasons, we stuck to the 10% unlearn budget, as this is a solid baseline used by the community.

**Why not more published MU methods?**: In terms of MU methods, it becomes impossible to include all published papers due to the growing number of publications in this space. We selected these six unlearning methods as they have been identified as the more promising methods by contemporary papers; as for the competition methods, while not published, we decided to evaluate them for two reasons: i) they all open-source code, 2) they have empirically shown in a competition that they were good performers. In this way, we aimed to establish a robust baseline of methodologies to extensively evaluate across model configurations and MU method hyperparameters and draw reliable conclusions on their performance.

**Extensions to other modalities and architectures**: Drawing concrete conclusions across modalities and model architectures is generally challenging, as each modality and architecture might have its properties. As such, drawing general conclusions would require novel experiments that are not included in this work. Therefore, we will update the limitations section of the article as follows.

**Limitations**:
This work focuses on vision data modalities using vision architectures for Image Classification. While computer vision represents a significant application area, other modalities and architectures should also be explored. We acknowledge that generalizing our results to other data types or models would indeed be beneficial. However, it is challenging to make broad extrapolations because different architectures may rely on distinct inductive biases, and different modalities or types of data often exhibit varying properties. For instance, in audio data and sequential data more generally, the interdependence and context across data points create additional complexities that make unlearning more difficult.

---

### Author Rebuttal · Authors · 2024-09-01

Again, we thank the reviewers for their feedback and suggestions.

Based on the discussion phase, here we list the minor revisions that we believe can be done in the camera-ready version:
- (1) To update the limitation section of the paper (Reviewer CWjQ).
- (2) To add a dedicated subsection explaining the distinction from the NeurIPS competition (Reviewers K9a2).
- (3) To add to the appendix, a more detailed table for comparing SOTA MU methods (Reviewer CWjQ).
- (4) To add to the appendix, a table summarizing the range of values within and across groups to facilitate comparison (Reviewer K9a2)
- (5) To better clarify the terminology and implementation setup, including the choice of epochs, samples of forget set, the ranking evaluation details, highlighting the reason for missing values, and addressing other inconsistencies in citations (Reviewers K9a2 and sLqq).
- (6) To organize and release the source code for reproducibility purposes (Reviewer T7vM).
- (7) To include one of the detailed-result tables from the appendix into the main body of the paper (Reviewer T7vM).

## Value for the community:

Our findings highlight the importance of comparing MU methods across diverse model seeds, datasets, and architectures. While some methods, like NG+ and FCS, generally perform well in many scenarios, they are not consistently the top performers across all settings.

Additionally, comparisons should be made under a fixed hyper-parameter tuning budget. Certain methods, like FCS, require extensive tuning over multiple hyper-parameters—such as optimizing the forget and retain sets with different optimizers and learning rates—whereas methods like CT or GA rely on a single optimizer, making their tuning process simpler.

## Updated Limitations:

This paper focuses exclusively on vision data modalities using vision architectures for image classification. While computer vision is a significant area of application, it is important to explore other modalities and architectures to generalize our findings. Different data types and models may rely on distinct inductive biases, and extrapolating our results to these other domains presents challenges. For example, in audio and sequential data, the interdependence and contextual relationships between data points introduce additional complexities, making the process of unlearning more difficult.

Additionally, this study was conducted with a fixed unlearning budget of 10% of the data. While it would be valuable to analyze other unlearning budgets, we prioritized a thorough investigation of this particular budget, which is already commonly used in the community.

Our analysis also focused on unlearning randomly selected data points rather than specific classes, features, or particularly challenging data. As a result, our findings may not fully generalize to these other scenarios. Nevertheless, we believe this setting can effectively represent actual deletion requests from users of an image classification service provider.

## Updated Distinction from the NeurIPS Unlearning Competition:

Our benchmark uses four datasets that are publicly available through PyTorch, along with one additional openly accessible dataset. Additionally, we evaluate multiple trained models initialized with different random seeds (10 seeds total), applying the MU methods to each of these models. This approach ensures that the best-performing methods demonstrate consistency across various initialization seeds.

In our evaluation, we included not only the top-performing methods from the competition but also methods from contemporary research papers. These additional methods were not part of the competition unless implemented by the participants, providing a broader comparison of performance.

Our findings indicate that the best-performing method from the competition is less reliable when tested across different architectures and datasets. Given that unlearning methods can be sensitive to hyper-parameter choices, our benchmark ensures fair comparisons by conducting a search of 300 hyper-parameter configurations (three searches of 100 combinations each), with each search starting from a model with different initial weights. In contrast, the competition allows for variations in this process across different submissions.

---

### Decision · Program_Chairs · 2024-09-26

**Decision:**

Reject

**Comment:**

This paper provides an evaluation of a number of machine unlearning methods, via a benchmark and evaluation study. With extensive hyperparameter searches, the authors find that two of the methods (MSG and CT) consistently perform better in terms of model accuracy and run time efficiency across different models and datasets. The authors also provide takeaways on the inadequacy of current baseline evaluation methods for MU.

The reviewers showed a wide range of responses to the paper, effectively identifying both positives and negatives of the paper, including:

pros:
- thoughtful evaluation of MU methods going beyond straight forward linear scores
- the work includes a fairly comprehensive list of MU methods, all carefully calibrated and experiments are well designed to capture a variety of useful metrics
- the paper is well written and easy to follow

cons:
- the main focus of the paper is on small classification benchmarks with limited number of classes
- the domain and model architectures are quite limited, and arguments for extensions and generalization beyond current architectures are not convincing

There are strong arguments for both positive and negative sides of this paper. The limitations of using only smaller architectures, for classification tasks, is very significant. There are no meaningful expectations that results from these smaller architectures would be indicative or even correlate with results on larger models today, multi-modal foundational models. For example, today's image classification tasks are now largely handled by multi-modal models like BLIP, LLaVA, Kosmos etc. Thus fixed class classification architectures will be extremely limited in relevance to real deployed applications, even on classification tasks.

While it may seem unfair to expect an academic paper to include results on massive foundational models, it is a challenge that is fundamental to benchmarking the large models that are most impactful in the wild today.